# EchoShot: Multi-Shot Portrait Video Generation

Jiahao Wang[1]    Hualian Sheng[2]    Sijia Cai[2,†]    Weizhan Zhang[1,*]    Caixia Yan[1]

Yachuang Feng[2]    Bing Deng[2]    Jieping Ye[2]

[1]School of Computer Science and Technology, MOEKLINNS, Xi'an Jiaotong University
[2]Alibaba Cloud Computing

jiahaowang@stu.xjtu.edu.cn   stephen.csj@alibaba-inc.com
zhangwzh@xjtu.edu.cn

## Abstract

Video diffusion models substantially boost the productivity of artistic workflows with high-quality portrait video generative capacity. However, prevailing pipelines are primarily constrained to single-shot creation, while real-world applications urge multiple shots with identity consistency and flexible content controllability. In this work, we propose EchoShot, a native and scalable multi-shot framework for portrait customization built upon a foundation video diffusion model. To start with, we propose shot-aware position embedding mechanisms within the video diffusion transformer architecture to model inter-shot variations and establish intricate correspondence between multi-shot visual content and their textual descriptions. This simple yet effective design enables direct training on multi-shot video data without introducing additional computational overhead. To facilitate model training within multi-shot scenarios, we construct PortraitGala, a large-scale and high-fidelity human-centric video dataset featuring cross-shot identity consistency and fine-grained captions such as facial attributes, outfits, and dynamic motions. To further enhance applicability, we extend EchoShot to perform reference image-based personalized multi-shot generation and long video synthesis with infinite shot counts. Extensive evaluations demonstrate that EchoShot achieves superior identity consistency as well as attribute-level controllability in multi-shot portrait video generation. Notably, the proposed framework demonstrates potential as a foundational paradigm for general multi-shot video modeling. Project page: https://johnneywang.github.io/EchoShot-webpage.

## 1 Introduction

Recent advancements in Diffusion Transformer-based (DiT) model [34] have catalyzed transformative progress in text-to-video (T2V) generation, enabling the synthesis of visually realistic single-shot videos [10, 6, 4, 7, 3, 43]. In real-world content creation workflows, users frequently demand multi-shot video generation with persistent subject consistency, particularly in human-centric applications such as narrative storytelling, virtual try-on with background variations, and appearance attribute editing. However, prevailing T2V models exhibit limitations in achieving identity consistency when generating multiple human-centric videos, attributable to pre-training on independently fragmented single-shot data and a lack of effective cross-shot post-training strategies.

To tackle this challenge, we present a novel multi-shot portrait video generation (MT2V) task. As straightforward solutions, existing studies can be organized to perform this task through two

---

† Project Lead
* Corresponding Author
  This work was completed during the internship at Alibaba Cloud Computing.

39th Conference on Neural Information Processing Systems (NeurIPS 2025).

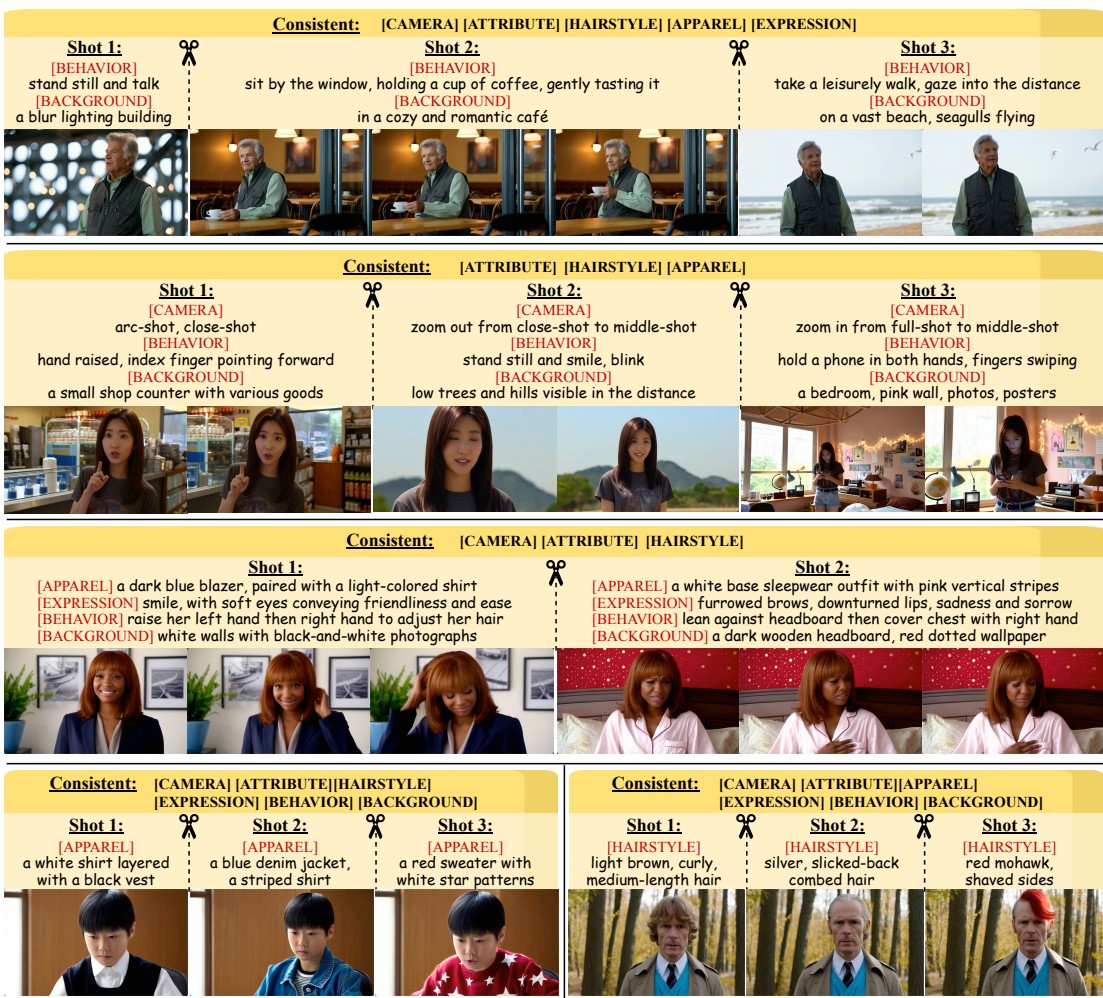

Figure 1: Given multiple formatted prompts of the same character, EchoShot generates multi-shot portrait videos showing the same appearance with superior fine-grained controllability.

paradigms: (1) Iterative personalized generation. This approach achieves cross-shot consistency through recurrent applications of personalized text-to-video (PT2V) models [7, 53, 4, 3] but faces challenges in terms of textual controllability and full-body consistency. To specify, the facial feature guidance mechanism in PT2V models often leads to adversarial competition between identity preservation and prompt adherence, reducing the controllability in aspects such as hairstyle, expressions, and apparel. Moreover, these models, tailored for facial preservation, fail to maintain the whole body consistency across generated videos. (2) Keyframe-to-video synthesis. This approach first generates multiple keyframes using consistent text-to-image (CT2I) models [56, 42, 20, 44, 45] and transforms them into multiple shots utilizing sophisticated image-to-video (I2V) models [4, 3, 43]. Although this decoupled strategy partially mitigates the multi-shot consistency challenge, it fundamentally suffers from the bottleneck effect and highly depends on the performance of CT2I models. The flaws of keyframes, such as blur, distortion, and inconsistency, will severely degrade the quality of generated videos. Besides, the absence of explicit identity conditioning in I2V models leads to progressive degradation of facial consistency during temporal expansion, causing dynamic identity drift.

For these issues, we explore a novel paradigm for native multi-shot consistent video generation in this work. Our motivation lies in the investigation of whether constructing a temporally coherent portrait dataset and implementing domain-specific post-training strategies can enable the DiT architecture to intrinsically model appearance-motion consistency across multi-shot videos. To this end, we introduce EchoShot, an advanced multi-shot portrait video generation framework built upon the state-of-the-art video DiT model Wan2.1-T2V [43], with our work contributing three core innovations: (1) We propose PortraitGala, a large-scale portrait dataset comprising 250k one-to-one single-shot video

clips and 400k many-to-one multi-shot video clips. Each video undergoes rigorous identity clustering via facial embeddings and is annotated with granular captions covering facial attributes, dynamic expressions, and scene conditions, ensuring strict consistency and fine-grained controllability in the final generated videos. (2) To enable efficient multi-shot post-training based on the pretrained T2V model, we integrate a new position embedding mechanism termed Temporal-Cut RoPE (TcRoPE) into the DiT-based visual token self-attention module to resolve challenges in variable shot counts, flexible shot durations, and inter-shot token discontinuities. In the video-text cross-attention module, a Temporal-Align RoPE (TaRoPE) with a mismatch suppression mechanism is further introduced to establish strict alignment between text descriptions and their corresponding video contents within each shot. (3) Based on our trained EchoShot model, we further extend its capabilities by incorporating an external facial encoder for reference image-based personalized multi-shot video generation (PMT2V), and designing a disentangled RefAttn mechanism for infinite shots video generation (InfT2V). These extensions highlight our model's flexibility and generalization in real-world content creation.

Our experiments demonstrate that EchoShot excels in generating multi-shot portrait videos with consistent cross-shot identity preservation, while supporting fine-grained attribute control and scene manipulation. Comprehensive evaluations confirm its superiority over existing methods in quantitative metrics and user studies, particularly in maintaining identity coherence under complex motions and content variations, all within a native multi-shot video generation architecture.

## 2   Related Work

**Video diffusion models**. The video generation field has witnessed remarkable progress over the past year, marked by Sora [10]'s pioneering adoption of a scalable DiT architecture. Subsequently, both closed-source models like Kling [4], MovieGen [35], Hailuo [3], Veo2 [6] and Gen-4 [2], along with open-source alternatives such as Open-Sora-Plan [27], CogVideoX [51], HunyuanVideo [24], StepVideo [31], and Wan [43], have achieved remarkable and rapid advancements in video generation quality. Key technical strides stem from architectural shifts from U-Net [36, 9] to DiT/MMDiT [34, 15], the evolution from spatio-temporal [38, 16] to 3D full attention in video token interactions, model optimization from DDPM [36] to flow matching [15], advanced Video-VAE [51, 43] and enhanced text encoders [35, 24]. Despite advancements, existing methods are limited to single-shot videos, and extending pretrained T2V models for multi-shot scenarios poses a critical challenge.

**Consistent generation**. Maintaining subject consistency across scenes has broad real-world applications, with early efforts focusing on image consistency for story visualization and comics. Techniques include iterative use of personalized text-to-image models [26, 47, 18] for similarity preservation, or leveraging self-attention mechanisms [56, 42] and in-context capabilities [20] for multi-image alignment. With advances in video generation, these approaches could be applied to keyframe generation, combined with I2V models for multi-shot video synthesis. For example, VideoStudio [30] integrates entity embeddings for appearance preservation, MovieDreamer [54] predicts coherent visual tokens that are then decoded into keyframes, and VGoT [55] employs identity-preserving embeddings for cross-shot consistency. However, challenges persist in achieving high-fidelity dynamic identity consistency and controllability. Recent work explores fine-tuning T2V models for multi-event and multi-shot generation. HunyuanVideo [24] supports two-shot generation through caption concatenation. TALC [8] enhances temporal alignment with improved text conditioning, MinT [48] introduces time-aware interactions and positional encoding, and LCT [17] extends single-shot models using scene-level attention, interleaved 3D embeddings, and asynchronous noise strategies. While LCT emphasizes cross-shot coherence via scripted entity descriptions, our method focuses on identity consistency and text-guided control in multi-shot portrait generation.

**Human video datasets**. Diverse, large-scale human-centric video datasets are crucial for advancing portrait video generation. While existing high-quality portrait datasets like CelebV-HQ [57], CelebV-Text [52], and VFHQ [49] primarily feature close-up heads or upper bodies, other action datasets such as UCF101 [39], ActivityNet [11], Kinetics700 [12] suffer from inadequate visual quality and inconsistent face visibility. Recently, OpenHumanVid [25] introduces a large-scale and visually realistic dataset sourced from films, TV series, and documentaries for single-shot video pretraining/fine-tuning. However, both specialized portrait datasets and general pre-training datasets [13, 46] currently only offer single-shot video clips, leaving a critical gap in high-quality multi-shot identity-consistent portrait video datasets.

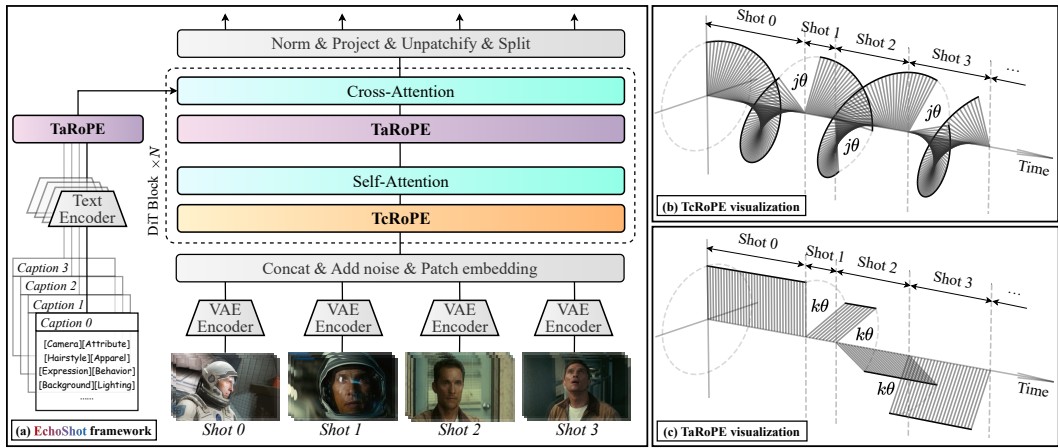

Figure 2: (a) The overall architecture of EchoShot, a multi-shot video generation paradigm, which features two intricate RoPE mechanisms. (b) TcRoPE, a 3D-RoPE that rotates an extra angular rotation at every inter-shot boundary along the time dimension. (c) TaRoPE, a 1D-RoPE that differentiates between matching and non-matching shot-caption pairs. Note that the visualization displays only one rotational component, with others excluded for simplicity.

# 3 Method

**Overview.** To accomplish multi-shot portrait video generation, we first review current single-shot models in Sec. 3.1. As a foundation, we devise and train a multi-shot text-to-video (MT2V) modeling paradigm in Sec. 3.2. We further tailor the personalized and infinite extensions in Sec. 3.3. All the training is driven by a meticulously curated dataset in Sec. 3.4.

## 3.1 Preliminary: Single-shot T2V Generation

Given a video segment $x$ during training, prevailing T2V models first encode it into a latent feature: $z = \mathcal{E}(x)$, where $\mathcal{E}$ is a pretrained video encoder. The latent feature is then mixed with Gaussian noise $\epsilon$, becoming a noisy sample $z_\tau$. The training is driven by a denoising process of rectified flow (RF) formulation [29, 15]: $\mathcal{L} = \mathbb{E}_{\epsilon_\tau, \tau, z} ||(\epsilon - z) - u_\phi(z_\tau, \tau, c)||^2$, where $\tau \in [0, 1]$ is the denoising timestep, $u_\phi(\cdot)$ is a denoising network and $c$ is the embeddings of the textual description. A typical implementation of $u_\phi(\cdot)$ is a DiT, combining self-attention and cross-attention layers. In the self-attention layers, given a 3D-position-indexed query or key vector $f^{sa}_{t,h,w} \in \mathbb{R}^d$, 3D Rotary Position Embedding (RoPE) [15] adds dimension-wise position information to it, forming $\tilde{f}^{sa}_{t,h,w}$:

$$\tilde{f}^{sa}_{t,h,w} = \text{3D-RoPE}(f^{sa}_{t,h,w}, t, h, w) = R^d_{\Theta_{3D},t,h,w} f^{sa}_{t,h,w}, \tag{1}$$

where $R^d_{\Theta_{3D},t,h,w}$ represents a 3D rotation matrix determined by the positional index and a set of base angles $\Theta_{3D}$. Details are provided in the Appendix. Notably, RoPE is generally excluded from cross-attention layers. While single-shot T2V models can generate portrait videos, extending this capability to multi-shot T2V models while preserving consistent identity remains a significant challenge.

## 3.2 Modeling Multi-shot T2V Generation for Consistent Identity

This task focuses on generating coherent multi-shot portrait videos that exhibit consistent identity across all shots while allowing flexible, user-defined control over both content and appearance. During training, each instance consists of a multi-shot video paired with corresponding frame counts and textual descriptions for each shot: $\{x_s, n_s, p_s\}_{s=0}^{S-1}$, where $S$ denotes the total number of shots. Here, $n_s$ represents the frame count for the $s$-th shot, and $p_s$ specifies its textual prompt. During inference, users are empowered to define the desired frame counts and prompts for $S'$ shots: $\{n_s, p_s\}_{s=0}^{S'-1}$, enabling customized multi-shot video generation with consistent identity. This task presents three key challenges compared to single-shot video generation:

- Variable Shot Lengths: Each shot's duration (i.e., frame count) can be flexibly defined by the user, accommodating diverse temporal requirements without imposing rigid constraints.
- Identity Consistency Across Shots: The generated video must ensure that all shots depict the same individual identity, maintaining seamless visual coherence throughout the sequence.
- Text-Driven Control of Each Shot: Both the background and foreground in each shot are controlled via textual prompts, enabling precise customization of scenes and characters.

**Identifying the inter-shot boundary via TcRoPE.** Starting with $S$ training video segments, a video encoder is first utilized to separately encode them into compressed latent features $z_s$, as shown in Fig. 2(a). Then, the latent features are further projected into query, key, and value embeddings to prepare for the self-attention mechanism. To preserve relative positional and temporal information, in vanilla 3D-RoPE, query and key embeddings require being modulated by the indexes of time, height, and width. However, such temporal index modulation is based on the assumption of continuous video content, which is not suitable for multi-shot scenarios with discontinuous video content. For example, the final frame of the first shot should correlate strongly with its preceding frame and weakly with the subsequent frame (i.e., the initial frame of the second shot). To model such an intricate correlation, as shown in Fig. 2(b), we raise TcRoPE to modulate the query and key embeddings, which incorporates a rotary phase shift between every two shots. For the feature embedding $a_{t,h,w,s}^{sa}$ at time $t$, height $h$, width $w$, and shot $s$, the processed feature can be denoted as:

$$\tilde{\boldsymbol{f}}_{t,h,w,s}^{sa} = \text{TcRoPE}(\boldsymbol{f}_{t,h,w,s}^{sa}, t, h, w, s) = \text{3D-RoPE}(\boldsymbol{f}_{t,h,w,s}^{sa}, t + s \cdot j, h, w), \qquad (2)$$

where $j$ is the phase shift scale. Intuitively, TcRoPE delineates the inter-shot boundary along the whole timeline concisely via an extra angular rotation. This method paves the way for precise control over the number of shots and the duration of each shot.

After the self-attention process, single-shot T2V models indiscriminately fuse semantic and visual modalities without incorporating positional information. In multi-shot scenarios, a straightforward approach is to concatenate all shot-wise captions into a single comprehensive caption. However, this method imposes limitations on the length of each shot-wise caption, given that the total caption must remain within the input window size of the text encoder.

**Associating the semantic and visual modality via TaRoPE.** Therefore, it is necessary to establish a shot-wise intricate correspondence when processing multi-shot videos and their associated captions. Specifically, the query of a specific shot should fully interact with the key of its corresponding caption, while maintaining a limited interaction with the keys of other captions, because simply ignoring interactions with other captions would risk discarding potentially useful supplementary details that could enhance the representation of the shot. To achieve such delicate interactions, we first input the captions separately into the text encoder to generate their respective embeddings. These embeddings are then concatenated to form the complete caption representation. Subsequently, as shown in Fig. 2(c), we propose TaRoPE to modulate the visual query embeddings and textual key embeddings. Given the $i$-th feature embedded in shot $s$, the processed feature can be expressed as:

$$\tilde{\boldsymbol{f}}_{i,s}^{ca} = \text{TaRoPE}(\boldsymbol{f}_{i,s}^{ca}, s) = \text{1D-RoPE}(\boldsymbol{f}_{i,s}^{ca}, s \cdot k), \qquad (3)$$

where $k$ is a hyperparameter that controls the mismatch suppression scale. Following feature modulation, attention calculation is performed, and the attention score between the query of $s_1$-th shot and the key of $s_2$-th shot is given by:

$$\tilde{A}_{s_1,s_2} = A_{s_1,s_2} \cdot \delta(k|s_1 - s_2|), \qquad (4)$$

where $A_{s_1,s_2}$ represents the standard vanilla attention, and $\delta(\cdot)$ is a monotonically decreasing function within a specific input interval, with $f(0) = 1$. Detailed derivation is provided in the Appendix. Consequently, the attention between the matched query and key (i.e., when $s_1 = s_2$) remains identical to that in vanilla models. In contrast, the attention between unmatched pairs is suppressed, controlled by $k$. Finally, the processed features are passed through an FFN to generate the output.

Our proposed TcRoPE supports variable shot lengths and counts, while TaRoPE enables text-driven control of each shot. The multi-shot training paradigm inherently ensures identity consistency across all shots. Notably, the two proposed RoPE mechanisms introduce no additional parameters.

### 3.3 Towards Personalized and Infinite Multi-shot T2V Generation

**PMT2V.** The aforementioned method enables the model to generate video segments while preserving consistent identity. However, a key application involves extending the model's capability to generate

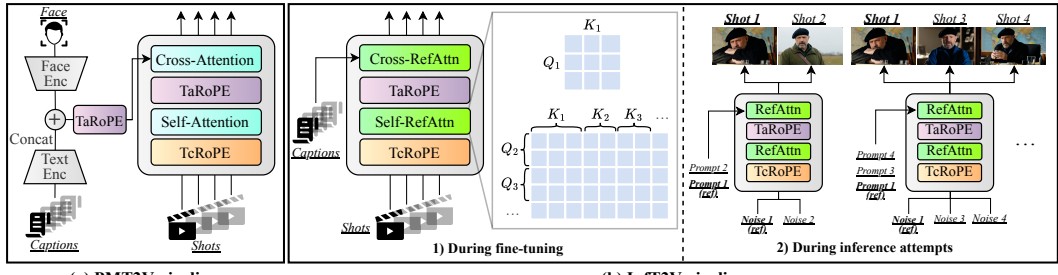

**(a) PMT2V pipeline**  **(b) InfT2V pipeline**

Figure 3: Two enhanced pipelines based on MT2V model. (a) PMT2V pipeline, with an integrated conditioner branch, generates multi-shot portrait videos of a given face input. (b) InfT2V pipeline creates infinite shots of the same person across multiple generation attempts, enabled by RefAttn, which disentangles the first shot as a constant reference.

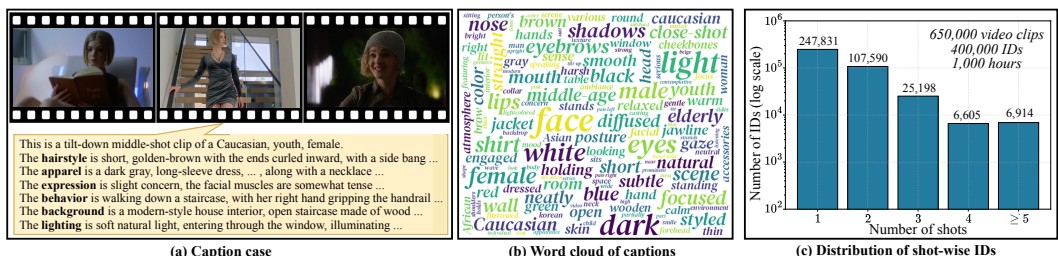

**(a) Caption case**  **(b) Word cloud of captions**  **(c) Distribution of shot-wise IDs**

Figure 4: (a) A caption case of PortraitGala. Each clip is thoroughly captioned in the fine-grained format. (b) The word cloud reflects the comprehensiveness of the captions. (c) PortraitGala consists of 650,000 clips with 400,000 IDs, totaling a video duration of 1,000 hours.

video segments corresponding to a user-specified image. To achieve this goal, as illustrated in Fig. 3(a), we incorporate a conditioning branch to encode the face input using a face encoder, which consists of ArcFace [14] and CLIP-G [41]. The resulting face embeddings are concatenated with the semantic embeddings of each text prompt, modulated by TaRoPE, and subsequently fed into the cross-attention layers. The denoising network is fine-tuned from the MT2V weights, guided by the standard RF loss.

**InfT2V.** In real-world applications, life-long video generation with an infinite number of shots for the same individual is required. A natural approach is to fix the first shot as a reference across multiple generation attempts while varying the remaining shots. The inherent dynamics of the full-attention mechanism update the denoising path of each shot based on the fixed noise input from the first shot and the random noise inputs from remaining shots, resulting in variations in the first shot across different generation attempts. To address this issue, we disentangle the first shot using RefAttn, as illustrated in Fig. 3, which computes two attention weight matrices:

$$A_1 = Q_1 \times K_1, \quad A_2 = [Q_2, \dots] \times [K_1, \dots], \tag{5}$$

where $[\cdot]$ is concatenation operation. Here, $A_1$ is employed to update the first shot, while $A_2$ is utilized to update the remaining shots. Integrating RefAttn, the model undergoes a quick fine-tuning to learn this new pattern. During inference, we maintain the noise, length, and prompt of the first shot unchanged across different generations, providing a fixed reference. As we change the prompts of the remaining shots, our method generates infinite shots while preserving the identity of the first.

### 3.4 PortraitGala: The First Multi-shot Portrait Video Dataset

**Data processing.** To support training, we meticulously construct a high-quality and large-scale human portrait dataset through an incremental process: 1) collect a raw data pool from diverse sources, including movies, episodes, open-source dataset, etc.; 2) filter by aspect ratio, resolution, and other criteria; 3) slice into single-shot clips using PySceneDetect [5]; 4) detect human face [23] and retain single-person clips; 5) identity many-to-one clips showing the same person using in-house

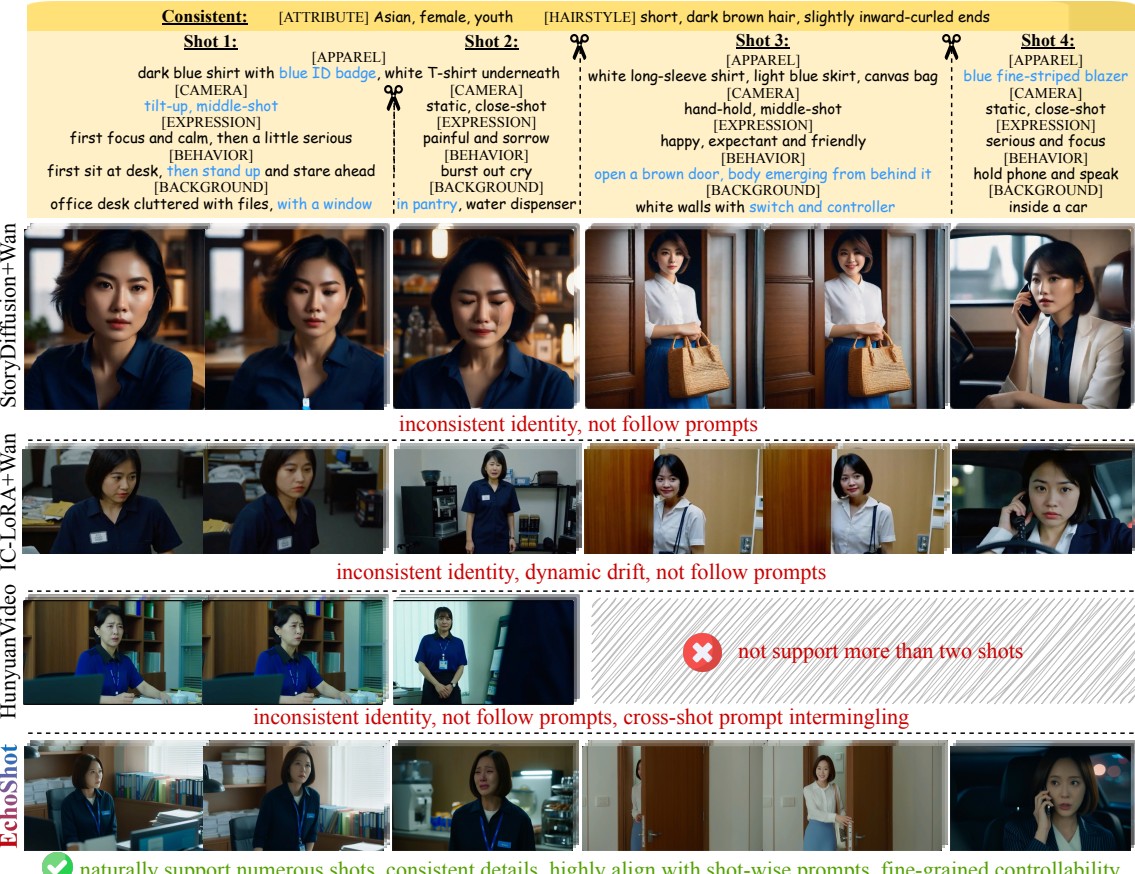

Figure 5: Illustration of EchoShot and baselines in the MT2V task. Key prompts are marked blue. Our method demonstrates superior appearance consistency and fine-grained controllability.

facial embedding extraction and clustering pipeline; 6) remove duplicate videos showing similar content; 7) attach attribute labels, text descriptions, and other necessary structured information based on Gemini 2.0 Flash [1].

**Data format and statistics.** We deconstruct the content of portrait videos into 8 aspects and establish a unified caption format. To specify, as shown in Fig. 4(a), every clip is annotated with the format "This is a [CAMERA] of a [ATTRIBUTE].[HAIRSTYLE].[APPAREL].[EXPRESSION].[BEHAVIOR].[BACKGROUND].[LIGHTING].". The word cloud in Fig. 4(b) reflects the comprehensiveness of our detailed captions, which endows our model with fine-grained controllability of the generated videos. As a result of the curation pipeline, we ultimately build up PortraitGala, which consists of 600k clips showing 400k identities, totaling a video duration of 1k hours. As depicted in Fig. 4(c), our dataset includes 247k single-shot IDs, 107k two-shot IDs, 25k three-shot IDs, and so on, which lays a solid foundation for our multi-shot training paradigm.

# 4 Experiments

**Implementation details.** EchoShot is implemented based on Wan-1.3B [43]. Throughout the three tasks, we set the resolution to $832 \times 480$ and use a fixed 125 frames, which is equivalent to 7.8 seconds in the real world. We set the phase shift scale $j$ to 4 and the mismatch suppression scale $k$ to 6. The training dataset consists of one-third one-to-one data and two-thirds many-to-one data, and the shot number $S$ varies from 1 to 4. The length of each shot is randomly sampled. All the training is driven by the standard RF loss. The MT2V pretraining takes about 3,500 NVIDIA A100 GPU hours. The PMT2V model and InfT2V model are fine-tuned based on the MT2V weights.

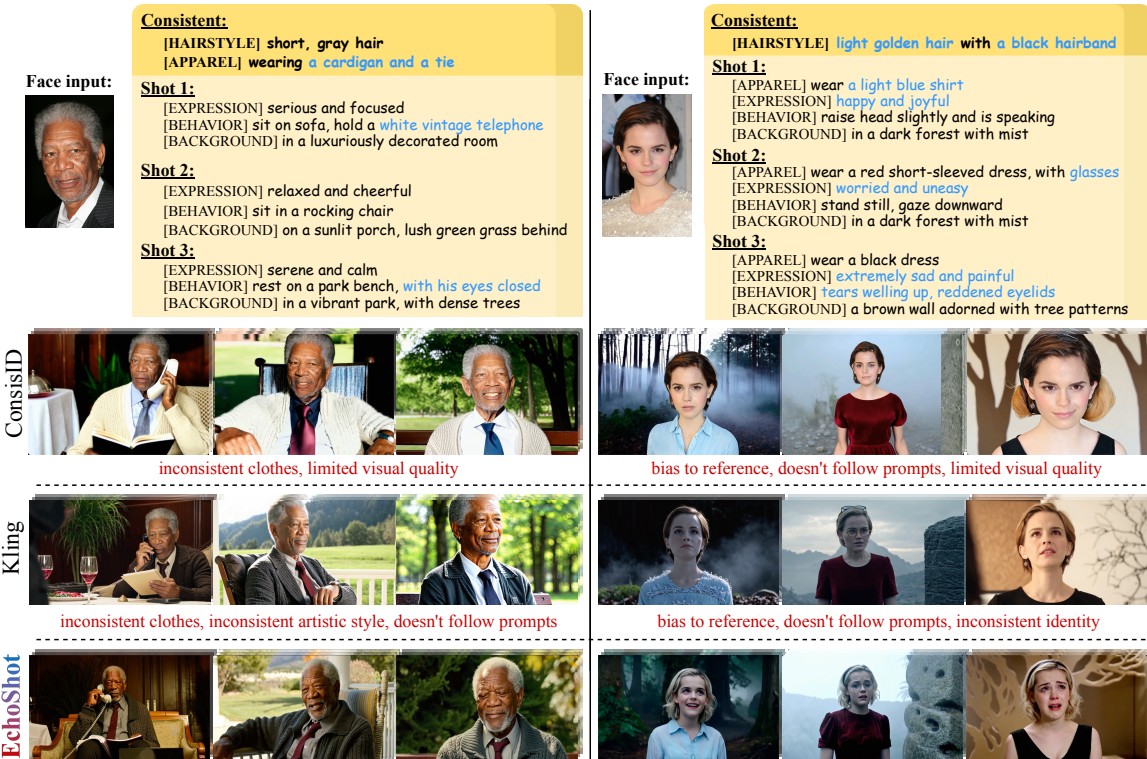

Figure 6: Illustration of EchoShot and baselines in PMT2V task. Baselines exhibit severe bias, poor consistency, and limited controllability, while our method demonstrates superior overall quality.

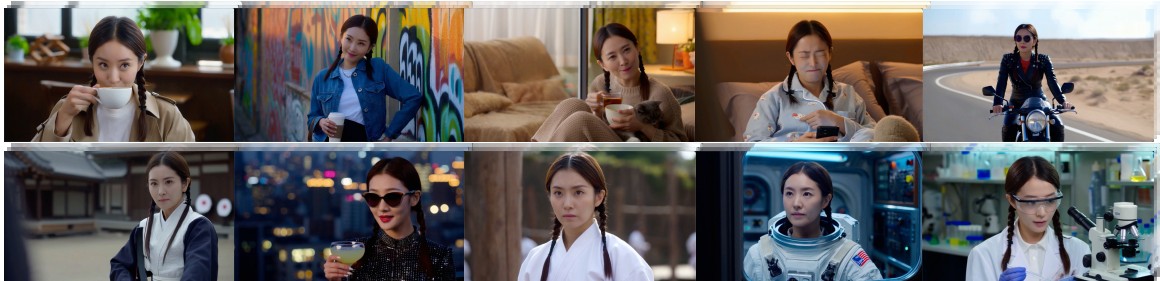

Figure 7: Illustration of EchoShot in InfT2V task. Our method succeeds in generating infinite shots (10 here) showing the same identity.

**Baselines.** To evaluate the performance of EchoShot, we compare it with a variety of baselines. For MT2V, we include (1) keyframe-based pipelines, StoryDiffusion [56]+Wan-I2V [43] (SD+W) and IC-LoRA [20]+Wan-I2V [43] (IC+W), (2) HunyuanVideo [24], which supports two shots generation. For PMT2V, we include (1) open-source ConsisID [53], (2) closed-source Kling [4]. All model settings are kept at their default values.

**Benchmark.** To provide an objective assessment, we instruct LLM to generate 100 sets of multi-shot prompts in the portrait caption format. The prompts are transformed into different versions to match the different text input lengths of baselines. We encompass metrics from three different aspects: (1) Identity consistency. Following [53], we use FaceSim-Arc [14] and FaceSim-Cur [21] to measure the facial consistency across different generated shots. (2) Prompt controllability. We instruct VLM to score the alignment between the video and the corresponding prompt in three dimensions: human appearance (App.), camera and human motions (Mot.), background and lighting (Bg.). (3) Visual quality. Since traditional metrics may not faithfully reflect human preferences, we follow [22] to instruct VLM to score in two dimensions. Static quality (Sta.) covers clarity, color saturation, content layout, etc. Dynamic quality (Dyn.) covers smoothness, temporal consistency, reasonableness, etc.

| Method | Identity consistency | | Prompt controllability | | | Visual quality | |
|---|---|---|---|---|---|---|---|
| | FaceSim-Arc | FaceSim-Cur | App. | Mot. | Bg. | Sta. | Dyn. |
| StoryDiffusion+Wan-14B | 68.65 | 65.29 | 83.33 | 72.50 | 69.53 | 63.20 | 84.68 |
| IC-LoRA+Wan-14B | 68.45 | 65.04 | 87.59 | 83.71 | 75.37 | 79.49 | **87.98** |
| HunyuanVideo-13B (2 shots) | 64.80 | 61.25 | 83.39 | 68.15 | 72.37 | 86.37 | 87.22 |
| EchoShot-1B (ours) | **73.74** | **69.43** | **95.84** | **88.86** | **94.72** | **88.91** | 87.85 |

Table 1: Metric results in MT2V task. Requiring only 1B parameters, EchoShot tops the scoreboard compared to baselines. The best and second-best scores are denoted **bold** and underlined.

## 4.1 Qualitative Evaluation

**MT2V.** As shown in Fig. 5, limitations of baselines are observed: (1) The performance of the keyframe-based pipelines is constrained by keyframe priors. To specify, SD+W fails to follow the detailed prompts (e.g., blue blazer, ID badge), and the visual style appears unnatural. Both SD+W and IC+W exhibit suboptimal consistency across shots. (2) The keyframe-based pipelines exhibit progressive degradation of facial consistency during temporal expansion. For example, the crying person in shot #2 generated by IC+W shows obvious inconsistency with other shots. (3) Simply concatenating the multi-shot prompts, adopted by HunyuanVideo, causes the intermingling of prompts of different shots and inferior controllability. For instance, in the videos generated by HunyuanVideo, the window intended for shot #1 unexpectedly appears in shots #2. By contrast, benefiting from the multi-shot modeling, our EchoShot generates numerous shots with consistent details as well as fine-grained controllability, demonstrating superiority over the baselines. Notably, EchoShot is able to generate a coherent sequence of human actions, such as the "body emerging from behind the door" in shot #2, whereas keyframe-based approaches struggle to achieve.

**PMT2V&InfT2V.** Fig. 6 illustrates the comparison between our method and baselines in PMT2V task. Primarily trained on a one-to-one portrait dataset, both ConsisID and Kling exhibit severe bias to the reference image and fail to follow the prompts (e.g., the hairstyle and expressions of Emma Watson). Besides, the repeated generation process lacks the ability to maintain consistency across different shots (e.g., the clothes of Morgan Freeman). By contrast, with many-to-one modeling and shot-aware mechanisms, our method achieves consistent appearance, vivid expressions, and precise controllability across shots. Additionally, we visualize the performance of our method in InfI2V task in Fig. 7. Thanks to the RefAttn mechanism, our method is able to generate infinite shots of freely customized prompts while maintaining the same identity of the character.

## 4.2 Quantitative Evaluation

**Metric results.** Tab. 1 shows the quantitative metrics of EchoShot and baselines in MT2V. For identity consistency, our method scores the highest 73.74% on FaceSim-Arc and 69.43% on FaceSim-Cur, achieving strong cross-shot identity consistency. In terms of prompt control, our method remarkably scores 95.84% App., 88.86% Mot., and 94.72% Bg., surpassing the baselines by a large margin. Besides, our method scores 88.91% Sta. and 87.85% Dyn., both top-tier. The results prove the overall advantages and the effectiveness of our method.

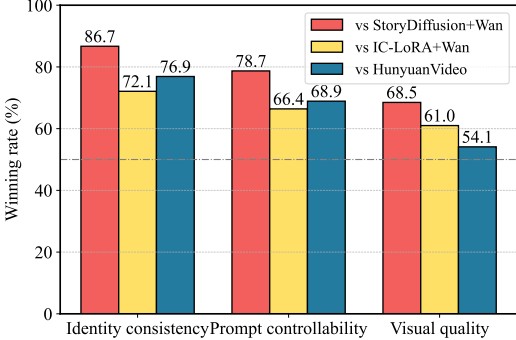

Figure 8: User study. The overall high winning rate proves that EchoShot better aligns with human preferences.

**User study.** To present a human-aligned assessment of MT2V, we further conduct a user study involving 20 participants. They are asked to perform binary voting in 45 one-on-one matchups between our method and baselines in terms of the above-mentioned three aspects. As shown in Fig. 8, our method wins at least 70% matchups for identity consistency and prompt controllability, as well as at least 50% matchups for visual quality. The user study reconfirms the superiority of our method, which echoes the metric results.

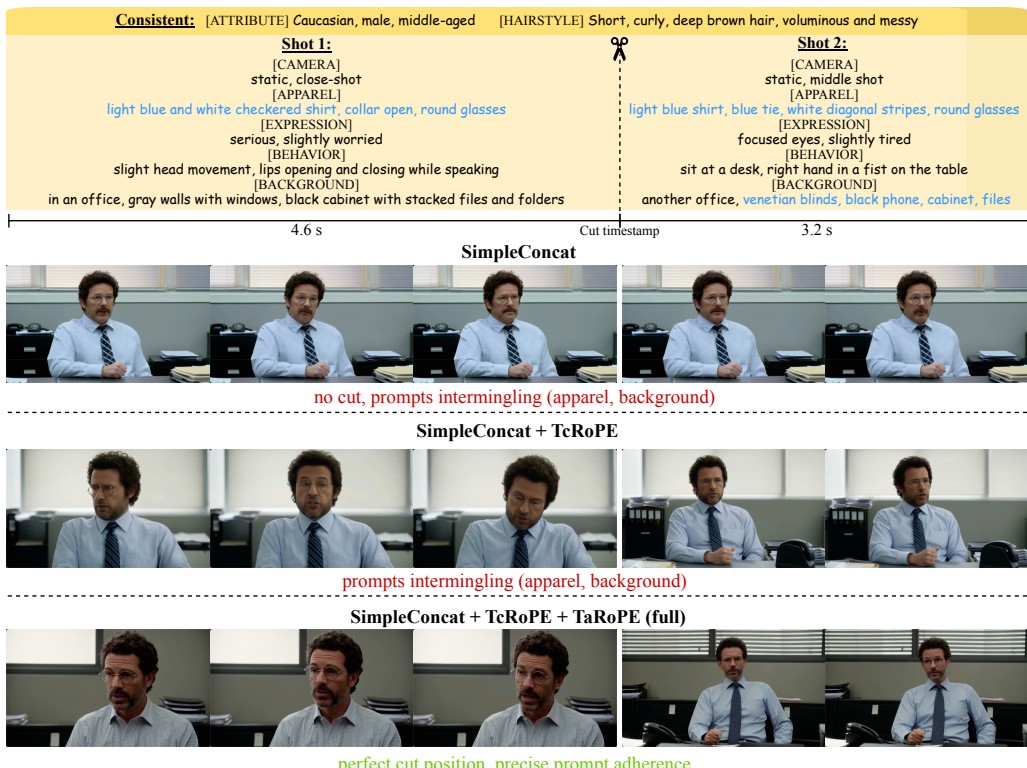

Figure 9: Illustration of generated videos of three ablation models, which confirms that TcRoPE and TaRoPE mechanisms perform as expected.

## 4.3 Ablation Study

To verify the effectiveness of each proposed mechanism, we carry out an ablation study. For rapid assessment, we build a reduced-scale dataset for ablation training. We construct three ablation models: (1) SimplyConcat (SC), which concatenates the videos and the captions from different shots indiscriminately, and applies vanilla RoPE. (2) SC+TcRoPE, which only applies the TcRoPE mechanism. (3) SC+TcRoPE+TaRoPE is our full method. As shown in Fig. 9, SC struggles to recognize the difference between shots, causing failure of shot transition. Though it occasionally generates multi-shot videos, the shot transition positions are elusive and uncontrolled. With the modeling of shot-wise boundaries, SC+TcRoPE enables shot transitions at given timestamps. Yet, both SC and SC+TcRoPE exhibit an intermingling effect between prompts from different shots (e.g., the clothes are the same across shots, though given different descriptions), likely caused by simple caption concatenation. In contrast, our full method demonstrates precise shot-wise prompt adherence without intermingling.

## 5 Conclusion

We introduce EchoShot, a novel framework for multi-shot portrait generation that addresses the limitations of existing single-shot pipelines. EchoShot achieves high-quality, identity-consistent multi-shot video synthesis with flexible user-defined control over content and appearance through two shot-aware RoPE mechanisms: TcRoPE and TaRoPE. Furthermore, we develop PortraitGala, a large-scale, high-fidelity human-centric video dataset designed to support cross-shot identity consistency and fine-grained controllability. Extensive experiments confirm that EchoShot surpasses existing methods in both quantitative metrics and qualitative assessments. Notably, EchoShot provides a promising solution to enhance artistic workflows in real-world applications.

*Limitation.* Our framework currently lacks the capability to directly extend shots from previous content, thereby limiting the generation of longer, continuous video segments within a single shot. Additionally, generating consistent multi-subject scripts is a promising direction for future work.

## Acknowledgements

This work was supported in part by NSFC under Grant 62192781, 62172326, 62137002, 62302384, Research Project Funded by the State Key Laboratory of Communication Content Cognition under Grant No. A202403, and the Project of China Knowledge Centre for Engineering Science and Technology.

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

# EchoShot: Multi-Shot Portrait Video Generation

## Appendix

## Table of Contents

# A Mathematical Formulation

## A.1 Derivation of Rotary Position Embedding

In Sec. 3.1, we introduce Rotary Position Embedding (RoPE) in vanilla models. Based on it, we propose two shot-aware mechanisms, TcRoPE and TaRoPE. Here, we provide the complete denotation of RoPE, following [40].

**1D-RoPE.** The proposed TaRoPE is based on 1D-RoPE, which is intended for sequential modality along a single dimension. Given a position-indexed query $\boldsymbol{q}_m \in \mathbb{R}^d$ or key vector $\boldsymbol{k}_m \in \mathbb{R}^d$, 1D-RoPE can be expressed as:

$$\tilde{\boldsymbol{q}}_m = \text{1D-RoPE}(\boldsymbol{q}_m, m) = \boldsymbol{R}^d_{\Theta_{1D}, m} \, \boldsymbol{q}_m, \tag{6}$$

$$\tilde{\boldsymbol{k}}_m = \text{1D-RoPE}(\boldsymbol{k}_m, m) = \boldsymbol{R}^d_{\Theta_{1D}, m} \, \boldsymbol{k}_m, \tag{7}$$

where $\boldsymbol{R}^d_{\Theta_{1D}, m}$ is the rotary matrix, denoted as:

$$\begin{pmatrix}
\cos m\theta_1 & -\sin m\theta_1 & 0 & 0 & \cdots & 0 & 0 \\
\sin m\theta_1 & \cos m\theta_1 & 0 & 0 & \cdots & 0 & 0 \\
0 & 0 & \cos m\theta_2 & -\sin m\theta_2 & \cdots & 0 & 0 \\
0 & 0 & \sin m\theta_2 & \cos m\theta_2 & \cdots & 0 & 0 \\
\vdots & \vdots & \vdots & \vdots & \ddots & \vdots & \vdots \\
0 & 0 & 0 & 0 & \cdots & \cos m\theta_{\frac{d}{2}} & -\sin m\theta_{\frac{d}{2}} \\
0 & 0 & 0 & 0 & \cdots & \sin m\theta_{\frac{d}{2}} & \cos m\theta_{\frac{d}{2}}
\end{pmatrix}, \tag{8}$$

with pre-defined parameters $\Theta_{1D} = \{\theta_i = 10000^{\frac{-2(i-1)}{d}}, i \in [1, 2, \cdots, \frac{d}{2}]\}$. Thus, the proposed TaRoPE in Eq. (3) can be further written as:

$$\tilde{\boldsymbol{q}}^{ca}_{i,s} = \text{TaRoPE}(\boldsymbol{q}^{ca}_{i,s}, s) = \text{1D-RoPE}(\boldsymbol{q}^{ca}_{i,s}, s \cdot k) = \boldsymbol{R}^d_{\Theta_{1D}, s \cdot k} \, \boldsymbol{q}^{ca}_{i,s}, \tag{9}$$

$$\tilde{\boldsymbol{k}}^{ca}_{i,s} = \text{TaRoPE}(\boldsymbol{k}^{ca}_{i,s}, s) = \text{1D-RoPE}(\boldsymbol{k}^{ca}_{i,s}, s \cdot k) = \boldsymbol{R}^d_{\Theta_{1D}, s \cdot k} \, \boldsymbol{k}^{ca}_{i,s}. \tag{10}$$

**3D-RoPE.** The proposed TcRoPE is based on 3D-RoPE, which is an extension of 1D-RoPE tailored for video modality. Given a 3D-position-indexed $\boldsymbol{q}_{t,h,w} \in \mathbb{R}^d$ or key vector $\boldsymbol{k}_{t,h,w} \in \mathbb{R}^d$, 3D-RoPE can be denoted as:

$$\tilde{\boldsymbol{q}}_{t,h,w} = \text{3D-RoPE}(\boldsymbol{q}_{t,h,w}, t, h, w) = \boldsymbol{R}^d_{\Theta_{3D}, t, h, w} \, \boldsymbol{q}_{t,h,w}, \tag{11}$$

$$\tilde{\boldsymbol{k}}_{t,h,w} = \text{3D-RoPE}(\boldsymbol{k}_{t,h,w}, t, h, w) = \boldsymbol{R}^d_{\Theta_{3D}, t, h, w} \, \boldsymbol{k}_{t,h,w}, \tag{12}$$

where $\boldsymbol{R}^d_{\Theta_{3D}, t, h, w}$ can be denoted as:

$$\begin{pmatrix}
\cos t\theta_1 & -\sin t\theta_1 & \cdots & 0 & 0 & 0 & 0 & \cdots & 0 & 0 & 0 & 0 & \cdots & 0 & 0 \\
\sin t\theta_1 & \cos t\theta_1 & \cdots & 0 & 0 & 0 & 0 & \cdots & 0 & 0 & 0 & 0 & \cdots & 0 & 0 \\
\vdots & \vdots & \ddots & \vdots & \vdots & \vdots & \vdots & \ddots & \vdots & \vdots & \vdots & \vdots & \ddots & \vdots & \vdots \\
0 & 0 & \cdots & \cos t\theta_{\frac{d}{6}} & -\sin t\theta_{\frac{d}{6}} & 0 & 0 & \cdots & 0 & 0 & 0 & 0 & \cdots & 0 & 0 \\
0 & 0 & \cdots & \sin t\theta_{\frac{d}{6}} & \cos t\theta_{\frac{d}{6}} & 0 & 0 & \cdots & 0 & 0 & 0 & 0 & \cdots & 0 & 0 \\
0 & 0 & \cdots & 0 & 0 & \cos h\theta_1 & -\sin h\theta_1 & \cdots & 0 & 0 & 0 & 0 & \cdots & 0 & 0 \\
0 & 0 & \cdots & 0 & 0 & \sin h\theta_1 & \cos h\theta_1 & \cdots & 0 & 0 & 0 & 0 & \cdots & 0 & 0 \\
\vdots & \vdots & \ddots & \vdots & \vdots & \vdots & \vdots & \ddots & \vdots & \vdots & \vdots & \vdots & \ddots & \vdots & \vdots \\
0 & 0 & \cdots & 0 & 0 & 0 & 0 & \cdots & \cos h\theta_{\frac{d}{6}} & -\sin h\theta_{\frac{d}{6}} & 0 & 0 & \cdots & 0 & 0 \\
0 & 0 & \cdots & 0 & 0 & 0 & 0 & \cdots & \sin h\theta_{\frac{d}{6}} & \cos h\theta_{\frac{d}{6}} & 0 & 0 & \cdots & 0 & 0 \\
0 & 0 & \cdots & 0 & 0 & 0 & 0 & \cdots & 0 & 0 & \cos w\theta_1 & -\sin w\theta_1 & \cdots & 0 & 0 \\
0 & 0 & \cdots & 0 & 0 & 0 & 0 & \cdots & 0 & 0 & \sin w\theta_1 & \cos w\theta_1 & \cdots & 0 & 0 \\
\vdots & \vdots & \ddots & \vdots & \vdots & \vdots & \vdots & \ddots & \vdots & \vdots & \vdots & \vdots & \ddots & \vdots & \vdots \\
0 & 0 & \cdots & 0 & 0 & 0 & 0 & \cdots & 0 & 0 & 0 & 0 & \cdots & \cos w\theta_{\frac{d}{6}} & -\sin w\theta_{\frac{d}{6}} \\
0 & 0 & \cdots & 0 & 0 & 0 & 0 & \cdots & 0 & 0 & 0 & 0 & \cdots & \sin w\theta_{\frac{d}{6}} & \cos w\theta_{\frac{d}{6}}
\end{pmatrix} \tag{13}$$

with pre-defined parameters $\Theta_{3D} = \{\theta_i = 10000^{\frac{-2(i-1)}{d}}, i \in [1, 2, \cdots, \frac{d}{6}]\}$. Thus, the proposed TcRoPE in Eq. (2) can be further written as:

$$\tilde{\boldsymbol{q}}^{sa}_{t,h,w,s} = \text{TcRoPE}(\boldsymbol{q}^{sa}_{t,h,w,s}, t, h, w, s) = \text{3D-RoPE}(\boldsymbol{q}^{sa}_{t,h,w}, t+s\cdot j, h, w) = \boldsymbol{R}^d_{\Theta_{3D}, t+s\cdot j, h, w} \, \boldsymbol{q}^{sa}_{t,h,w,s}, \tag{14}$$

$$\tilde{\boldsymbol{k}}^{sa}_{t,h,w,s} = \text{TcRoPE}(\boldsymbol{k}^{sa}_{t,h,w,s}, t, h, w, s) = \text{3D-RoPE}(\boldsymbol{k}^{sa}_{t,h,w}, t+s\cdot j, h, w) = \boldsymbol{R}^d_{\Theta_{3D}, t+s\cdot j, h, w} \, \boldsymbol{k}^{sa}_{t,h,w,s}. \tag{15}$$

## A.2 Proof of Claim about Attention Score with TaRoPE

In Eq. (4), we give a concise formulation of the attention score after TaRoPE with claims. We provide a detailed proof of the claim here. Given queries of the $s_1$-th shot $\boldsymbol{q}_{s_1}$ and keys of the $s_2$-th shot $\boldsymbol{k}_{s_2}$, if we divide them into 2-component pairs along the channel dimension, the inner product of their TaRoPE-enhanced representations can be written as a complex number multiplication:

$$
\begin{aligned}
\tilde{\boldsymbol{A}}_{s_1,s_2} &= (\boldsymbol{R}^d_{\Theta,ks_1}\boldsymbol{q}_{s_1})^\top (\boldsymbol{R}^d_{\Theta,ks_2}\boldsymbol{k}_{s_2}) \\
&= \mathrm{Re}\Bigg[\sum_{i=0}^{\frac{d}{2}-1} \boldsymbol{q}_{s_1,[2i:2i+1]}\boldsymbol{k}^*_{s_2,[2i:2i+1]}e^{ik(s_1-s_2)\theta_i}\Bigg] \\
&= \sum_{i=0}^{\frac{d}{2}-1}(q_{s_1,2i}\,k_{s_2,2i} + q_{s_1,2i+1}\,k_{s_2,2i+1})\cos\left(k(s_1-s_2)\theta_i\right) \\
&\qquad + (q_{s_1,2i}\,k_{s_2,2i} - q_{s_1,2i+1}\,k_{s_2,2i+1})\sin\left(k(s_1-s_2)\theta_i\right),
\end{aligned}
\tag{16}
$$

where $\boldsymbol{k}_{s_1,[2i:2i+1]}$ represents the $2i^{th}$ to $(2i+1)^{th}$ components of $\boldsymbol{k}_{s_1}$ and $k_{s_1,2i}$ represents the $2i^{th}$ component of $\boldsymbol{k}_{s_1}$. Note that the scalar $k$ without subscripts is the mismatch suppression scale. When $\boldsymbol{q}_{s_1}$ and $\boldsymbol{k}_{s_2}$ are from the same shot (i.e., $s_1 = s_2$), Eq. (16) can be simplified as:

$$
\tilde{\boldsymbol{A}}_{s_1,s_2} = \sum_{i=0}^{\frac{d}{2}-1}(q_{s_1,2i}\,k_{s_2,2i} + q_{s_1,2i+1}\,k_{s_2,2i+1}) = \boldsymbol{q}_{s_1}\,\boldsymbol{k}_{s_2} = \boldsymbol{A}_{s_1,s_2},
\tag{17}
$$

which indicates that the attention score with TaRoPE remains exactly the same as that without TaRoPE.

To investigate the behavior of Eq. (16) as $k(s_1 - s_2)$ varies, we follow [40] to denote $\boldsymbol{h}_i = \boldsymbol{q}_{[2i:2i+1]}\boldsymbol{k}^*_{[2i:2i+1]}$, $S_j = \sum_{i=0}^{j-1}e^{i(s_1-s_2)\theta_i}$, and let $h_{\frac{d}{2}} = 0$, $S_0 = 0$, we can rewrite Eq. (16) using Abel transformation:

$$
\sum_{i=0}^{\frac{d}{2}-1}\boldsymbol{q}_{[2i:2i+1]}\boldsymbol{k}^*_{[2i:2i+1]}e^{i(s_1-s_2)\theta_i} = \sum_{i=0}^{\frac{d}{2}-1}h_i(S_{i+1}-S_i) = -\sum_{i=0}^{\frac{d}{2}-1}S_{i+1}(h_{i+1}-h_i).
\tag{18}
$$

$$
\begin{aligned}
|\sum_{i=0}^{\frac{d}{2}-1}\boldsymbol{q}_{[2i:2i+1]}\boldsymbol{k}^*_{[2i:2i+1]}e^{i(s_1-s_2)\theta_i}| &= |\sum_{i=0}^{\frac{d}{2}-1}S_{i+1}(h_{i+1}-h_i)| \\
&\leq \sum_{i=0}^{\frac{d}{2}-1}|S_{i+1}||(h_{i+1}-h_i)| \\
&\leq \left(\max_i|h_{i+1}-h_i|\right)\sum_{i=0}^{\frac{d}{2}-1}|S_{i+1}| \\
&\leq \sum_{i=0}^{\frac{d}{2}-1}|S_{i+1}|
\end{aligned}
\tag{19}
$$

We plot the graph of the value of $f(x) = \sum_{i=0}^{\frac{d}{2}-1}|S_{i+1}|$ as $x = k(s_1 - s_2)$ varies in [0,50] in Fig. 10, which shows a monotonically decreasing function. By synthesizing the above derivations, we arrive at the conclusion of Eq. (4).

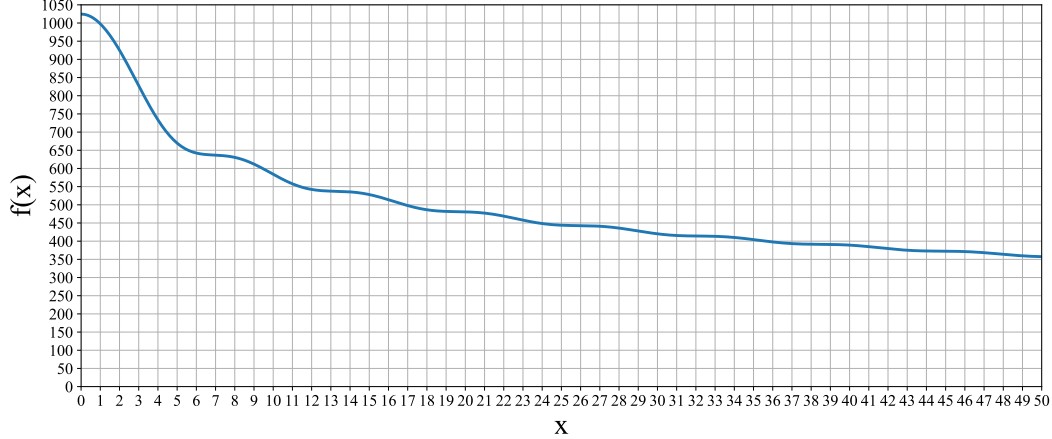

Figure 10: Graph of the attention score $f(x)$ as $x$ varies.

# B    Details of PortraitGala

## B.1    Data Processing

We detail the construction of the PortraitGala dataset here, adhering to the methodology outlined in Sec. 3.4. The PortraitGala dataset comprises a collection of video resources, drawing from three primary sources: 1) publicly available, high-quality portrait datasets such as CelebV-HQ [57] and CelebV-Text [52]; 2) subsets extracted from large-scale, open-source video datasets including OpenVid-1M [33] and OpenHumanVid [25]; and 3) a selection of videos obtained from various websites, encompassing both films and television series. In its entirety, the original dataset encompassed 26,967 hours of video footage.

The curation process began with rigorous filtering of raw video material to ensure quality and relevance. Specifically, we eliminated videos exhibiting vertical aspect ratios or substandard resolution, retaining only those conforming to the criteria: width > height > 480 pixels. Given the extended duration of many source videos and the presence of multiple camera shots within them, we employed PySceneDetect [5] to segment these videos into discrete, single-shot clips. Furthermore, to maintain high aesthetic quality, clips with an aesthetic score, as determined by [37], below a threshold of 4 were discarded.

To ensure that each video clip in PortraitGala featured only one individual, we implemented a person counting procedure leveraging YOLOv11 [23] for person detection and tracking. Subsequently, person identities were assigned using an improved version of HDBSCAN [32], a density-based clustering algorithm, coupled with a proprietary facial embedding extraction technique. A remaining challenge was the identification and removal of near-duplicate video clips depicting the same individual performing similar actions within a consistent setting. To address this, we simply employed Non-Maximum Suppression based on the detected person bounding boxes, specifically for videos with matching person IDs. Through rigorous manual verification, we achieve a clustering accuracy exceeding 99%, demonstrating that all video clips assigned identical identifiers consistently depict the same individual throughout.

Finally, attributes pertaining to camera shot composition and shot type were derived using internal analytical methods. Supplementary attribute labels and descriptive text were generated with the aid of Gemini 2.0 Flash [1] using the prompt below.

---

1. You are now going to describe the film for audience. The description should be as detailed, comprehensive, and logical as possible.

2. The output must be in English, around 600 words.

3. Do not use vague words such as "appear to be" or "seem." Your descriptions must be definitive, precise, and highly accurate.

4. There is only one main person, and all descriptions revolve around this person.

5. All sentences must be affirmative statements. No questions or negative sentences are allowed.

6. Do not determine the gender of the main character. Use "the person," "the person is," or "the person's" as pronouns.

7. Please provide a description of the entire video, rather than describing each individual image.

8. Each input requires descriptions of the following eight aspects, with each aspect not exceeding 100 words, and the key points of the descriptions between the aspects should not overlap:

   (a) **Hairstyle:** Provide a detailed and comprehensive description of the hairstyle of the person. Start with 'The person's hair ...'.

   (b) **Expression:** Provide a detailed and comprehensive description of the expressions and emotions of the person. (If there are any temporal changes in expression, also need describe in detail.) Start with 'The person's expression ...'.

   (c) **Apparel:** Provide a detailed and comprehensive description of the clothing worn by the person (including lower body garments if visible). If there are accessories (e.g., glasses, earrings, watches), describe them as well. Start with 'The person wears ...'.

   (d) **Behavior:** Provide a detailed and comprehensive description of the behavior of the person (e.g., actions, interactions with objects). Start with 'The person ...'.

   (e) **Background:** From the perspective of film set design, provide a detailed and comprehensive description of the background and environment. Start with 'The background ...'.

   (f) **Lighting:** From the perspective of film analysis, provide a detailed and comprehensive description of the overall lighting conditions. 'The lighting ...'.

## B.2  Data Distribution

We perform a statistical analysis on some key aspects of PortraitGala dataset, depicted in Fig. 11. The shot-scale distribution indicates that most videos are middle-shot (37.8%), followed by close-shot (33.6%), long-shot (15.9%), and full-shot (12.7%). In terms of gender distribution, male individuals are predominant at 56.7%, while female individuals constitute 39.7%, and others make up 3.6%. The age-group distribution shows a vast majority of youth individuals at 74.9%, with middle-aged individuals at 22.6%, and other age groups at 2.5%. Finally, ethnicity distribution reveals that 51.2% of the shown individuals are white people, 19.3% are black people, 11.4% are Asian people, and 18.1% are other ethnicities. This statistic breakdown highlights the diversity present in the PortraitGala dataset.

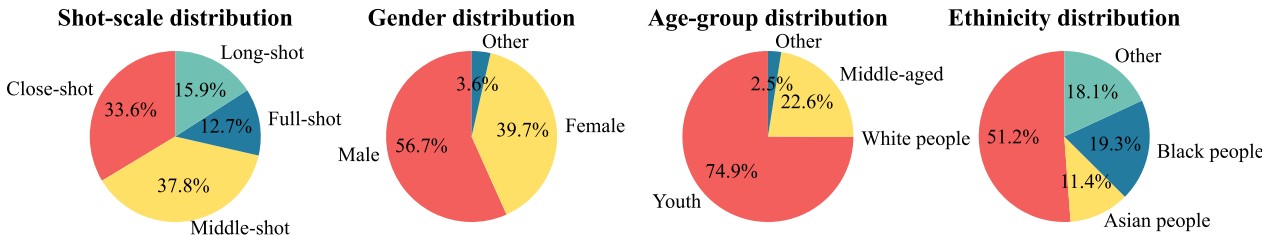

Figure 11: Distribution of key aspects of PortraitGala.

## B.3  Dataset Examples

To provide a clear and intuitive representation of the dataset, we present a gallery of multi-shot video instances in Fig. 12. It highlights the diversity of the video distribution and the granularity of the captions.

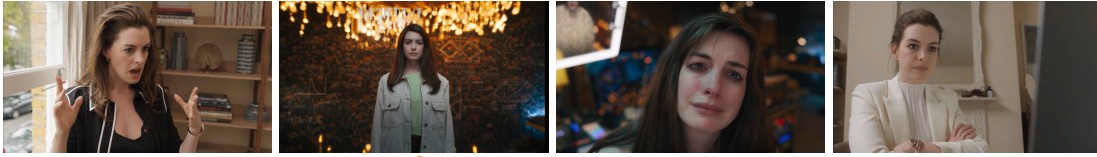

1. This is a zoom-out middle-shot clip of a Caucasian, youth, female.
2. The **hairstyle** is long, straight, and dark brown. It is parted centrally and cascades smoothly over both shoulders, extending past the chest.
3. The **apparel** is a light beige corduroy jacket with a collar, button front, and two flap chest pockets. Underneath, a light green t-shirt is visible, featuring a red trim around its crew neckline.
4. The **expression** is consistently neutral and composed. The gaze is direct towards the camera with steady, open eyes. The mouth remains closed, lips slightly pressed together, conveying a serious and focused demeanor throughout the clip.
5. The **behavior** is standing perfectly still and upright in the center of the frame. There are no discernible movements of the body or limbs. The person maintains an unwavering direct gaze at the camera, without any interaction with objects.
6. The **background** is a large, dark chalkboard densely covered with white chalk. It displays complex mathematical equations, scientific graphs like a normal distribution curve, and various geometric patterns. The setting signifies an academic or scientific environment.
7. The **lighting** is warm, golden light emanates from a dense array of exposed light bulbs positioned above. A direct key light illuminates the person's face from the front. The overall lighting is bright and focused on the person, with the background also visibly lit.

Figure 12: Illustration of examples in PortraitGala.

# C More Experimental Details

All the training is carried out on NVIDIA A100 80GB GPUs. The MT2V pretraining takes 3,500 GPU hours while the PMT2V and InfT2V take additional 2000 and 1000 GPU hours, respectively. Throughout the training, all the important settings are listed below:

| Parameter | Value |
|---|---|
| Video height | 480 |
| Video width | 832 |
| Video frame | 125 |
| FPS | 16 |
| Batchsize | 2 |
| Train timesteps | 1000 |
| Train shift | 5.0 |
| Optimizer | AdamW |
| Learning rate | 8e-6 |
| Weight decay | 0.001 |
| Sample timesteps | 50 |
| Sample shift | 5.0 |
| Sample guidance scale | 5.0 |

Table 2: Experimental settings of EchoShot.

# D Extra Experiments

## D.1 Motivation Verification

The pivotal core of our method is the multi-shot text-to-video modeling with the proposed two mechanisms, TcRoPE and TaRoPE. To affirm our motivation as well as verify the effectiveness, we conduct an intuitive experiment between EchoShot and the vanilla model. To specify, we perform a three-shot portrait video generation with EchoShot given three cut timestamps and three prompts. For the vanilla model, we concatenate the three prompts together as a input and perform standard generation. During generation, we collect the self-attention scores and cross-attention scores of the DiT blocks, average them and plot the heatmaps. As shown in Figs. 13(a) and (b), the vanilla model indiscriminately allocates attention to the visual tokens and the textual tokens, which is suitable for single-shot video modeling. Yet in the context of multiple shots, the vanilla model fails to differentiate the shots. By contrast, as shown in Fig. 13(c), TcRoPE establishes a inter-shot boundary during self-attention calculation, with the extra rotation naturally reduce the interactions of features from different shots while enhance that from the same shot. Additionally, TcRoPE allows the model to flexibly allocate attention to potentially crucial pixels, whether from the same shot or from different shots. Fig. 13(d) confirms a similar conclusion to the above one, in the context of TaRoPE and cross-attention.

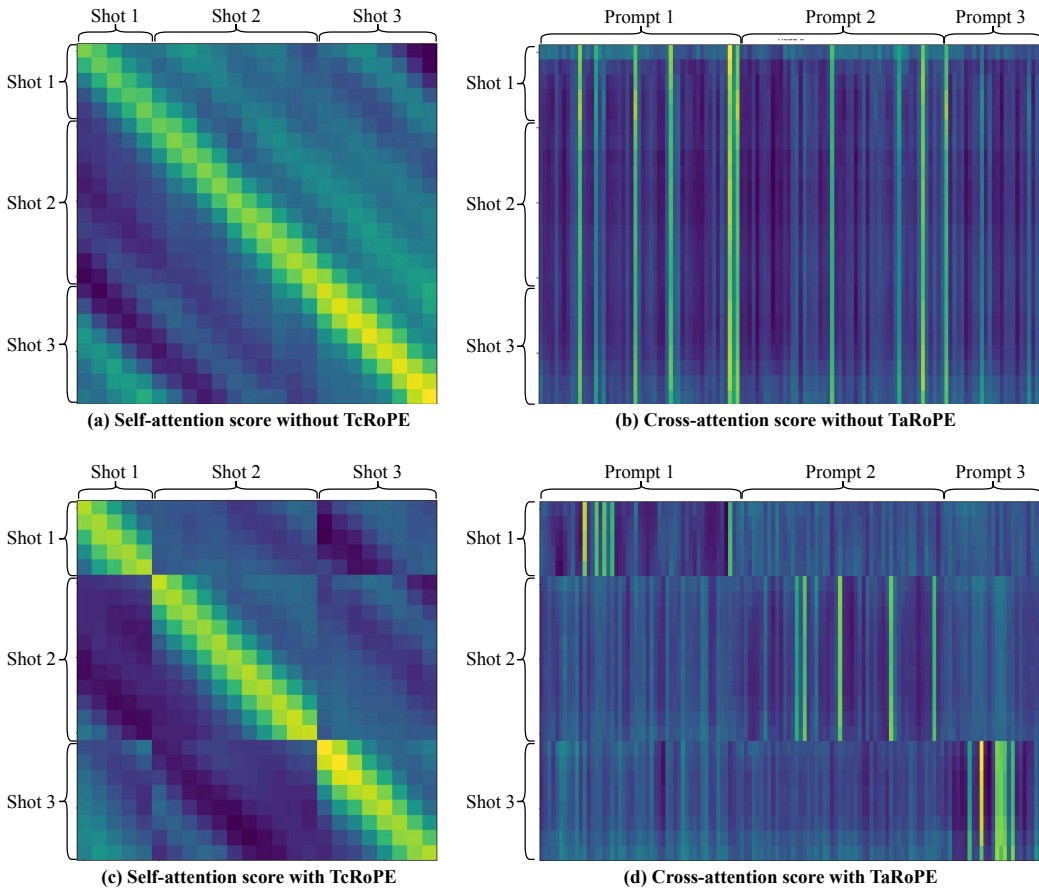

Figure 13: Visualization of self-attention score matrix w/ and w/o TcRoPE and cross-attention score matrix w/ and w/o TaRoPE.

## D.2 More Quantitative Results

In Tab. 1, we present the quantitative results. As a supplement, we propose the stand deviation results of these metrics across the test set here.

| Method | Identity consistency | | Prompt controllability | | | Visual quality | |
|---|---|---|---|---|---|---|---|
| | FaceSim-Arc | FaceSim-Cur | App. | Mot. | Bg. | Sta. | Dyn. |
| StoryDiffusion+Wan-14B | 68.65 | 65.29 | 83.33 | 72.50 | 69.53 | 63.20 | 84.68 |
| IC-LoRA+Wan-14B | 68.45 | 65.04 | 87.59 | 83.71 | 75.37 | 79.49 | **87.98** |
| HunyuanVideo-13B (2 shots) | 64.80 | 61.25 | 83.39 | 68.15 | 72.37 | 86.37 | 87.22 |
| EchoShot-1B (ours) | **73.74** | **69.43** | **95.84** | **88.86** | **94.72** | **88.91** | 87.85 |

| Method | Identity consistency | | Prompt controllability | | | Visual quality | |
|---|---|---|---|---|---|---|---|
| | FaceSim-Arc | FaceSim-Cur | App. | Mot. | Bg. | Sta. | Dyn. |
| StoryDiffusion+Wan-14B | 6.24 | 6.86 | 9.50 | 22.47 | 21.29 | 15.20 | 2.57 |
| IC-LoRA+Wan-14B | 7.02 | 6.75 | 10.16 | 18.41 | 19.77 | 13.08 | 2.68 |
| HunyuanVideo-13B (2 shots) | 9.16 | 8.28 | 10.13 | 18.35 | 17.95 | 5.76 | 3.51 |
| EchoShot-1B (ours) | 7.13 | 7.91 | 6.51 | 11.25 | 6.68 | 4.90 | 2.77 |

Table 3: Metric results (mean and standard deviation) in MT2V task.

| Method | Cut control | Identity consistency | | Prompt controllability | | | Visual quality | |
|---|---|---|---|---|---|---|---|---|
| | | FaceSim-Arc | FaceSim-Cur | App. | Mot. | Bg. | Sta. | Dyn. |
| SC | No | 75.24 | 70.61 | 55.79 | 59.16 | 47.56 | 81.90 | 75.24 |
| SC+TcRoPE | Yes | 74.77 | 69.85 | 81.72 | 72.84 | 74.22 | 84.32 | 81.69 |
| SC+TcRoPE+TaRoPE(full) | Yes | 75.83 | 70.58 | 94.12 | 87.41 | 93.96 | 84.10 | 83.46 |

Table 4: Metric results of three ablation models on reduced-scale dataset.

## D.3 Quantitative Ablation Study

We further conduct a quantitative ablation study following Sec. 4.3 in Tab. 4. Benefiting from the curated high-quality multi-shot dataset, the identity consistency and visual quality of three ablation models exhibit minimal variation. In terms of prompt controllability, the full model demonstrates significant advantages over the comparing models, echoing the qualitative results. The ablation study confirms that TcRoPE and TaRoPE mechanisms perform as expected.

## D.4 Parameter Analysis

Our method relies primarily on two key hyperparameters, the phase shift scale $j$ in Eq. (2) and the mismatch suppression scale $k$ in Eq. (3). To investigate the appropriate parameter selection, we conduct an intuitive parameter analysis with the reduced-scale dataset. We leverage the mathematical analysis and empirically pick several typical values, $j \in \{2, 4, 6\}, k = 6$ and $j = 4, k \in \{2, 6, 12\}$. The quantitative results of the models under different settings are shown in Tab. 5. To summarize, a too small $j$ excessively enhances the inter-shot visual interactions, leading to occasional cut failure and prompt intermingling with low prompt controllability scores. While a too large $j$ nearly blocks the interactions, causing identity inconsistency with low identity consistency scores. Similarly, a too small $k$ gives rise to prompt intermingling while a too large $k$ results in a slightly decrease in scores. Consolidating the results, we choose $j = 4, k = 6$ as the default setting.

| Method | Identity consistency | | Prompt controllability | | | Visual quality | |
|---|---|---|---|---|---|---|---|
| | FaceSim-Arc | FaceSim-Cur | App. | Mot. | Bg. | Sta. | Dyn. |
| j=2,k=6 | 75.45 | 70.64 | 88.75 | 82.10 | 85.28 | 80.97 | 81.67 |
| j=6,k=6 | 70.19 | 67.30 | 94.25 | 88.29 | 92.66 | 83.18 | 82.03 |
| j=4,k=2 | 75.12 | 69.73 | 84.20 | 76.69 | 80.16 | 80.55 | 81.02 |
| j=4,k=12 | 74.98 | 69.45 | 92.15 | 86.82 | 92.73 | 83.50 | 83.91 |
| j=4,k=6(ours) | 75.83 | 70.58 | 94.12 | 87.41 | 93.96 | 84.10 | 83.46 |

Table 5: Metric results of different parameter settings on reduced-scale dataset.

# E    EchoShot Gallery

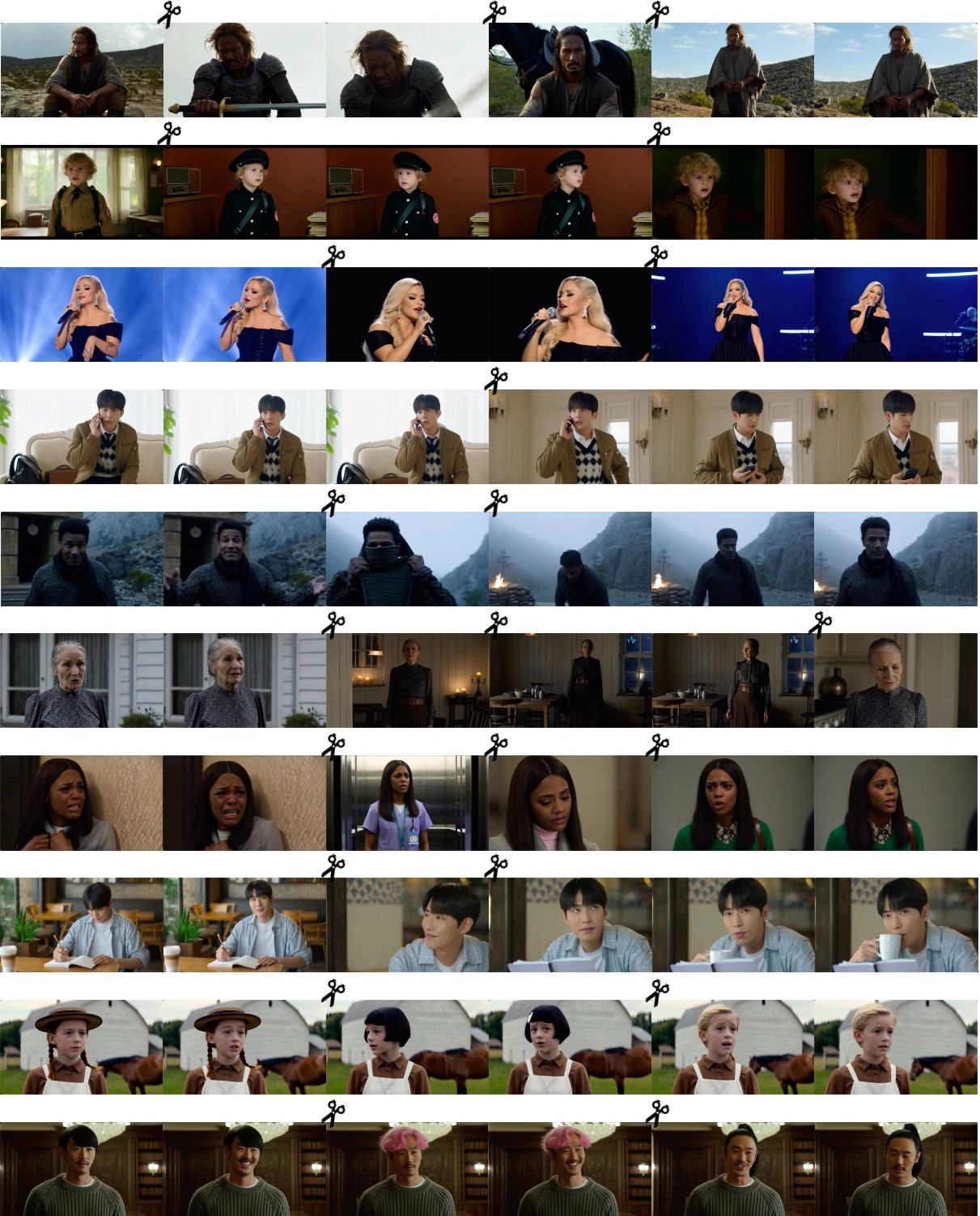

Figure 14: Illustration of generated videos by EchoShot.

# F  Clarification about User Study

**Instructions for Participants**

---

**Thank you for participating in our study!**

We appreciate your willingness to contribute to our research. Below, you will find detailed instructions on how to complete the tasks involved in this study. Please read this information carefully before beginning.

**1. Purpose of the Study**

The goal of this study is to evaluate different methods for generating multi-shot portrait videos showing the same identity. Your feedback will help us evaluate which method produce videos with best quality. Your participation is highly valuable to us!

**2. Task Overview**

You will be asked to compare two methods by evaluating their outputs in a series of one-on-one matchups. Each matchup will present results generated by our proposed method and a random baseline method (both anonymous). Your task is to vote for your preferred option in each matchup based on the following criteria:

- **Identity Consistency**: Which method better preserves the same identity of the human (e.g., facial features, expressions, and overall appearance) across all frames and shots in the video?
- **Prompt Controllability**: Which method more accurately follows the given prompts (e.g., appearance, motions, expressions, and background) in the generated video?
- **Visual Quality**: Which method produces a video with higher visual quality, considering factors like sharpness, smoothness of motion, and absence of artifacts (e.g., blurriness or unnatural textures)?

**3. How to Complete the Task**

- You will complete **45 matchups** in total.
- For each matchup, you will see two outputs side by side.
- To cast your vote, click on the button corresponding to your preferred option.
- There are no right or wrong answers. Please choose the option that feels best to you.

**4. Time Commitment**

The study should take approximately **20-30 minutes** to complete. You can work at your own pace, but we recommend completing the task in one sitting to ensure consistency.

**5. Voluntary Participation**

Your participation in this study is entirely **voluntary**. You are free to withdraw at any time without penalty or explanation. If you decide to withdraw, your responses will not be included in the analysis.

**6. Compensation**

This study does not provide financial compensation. However, your contribution is greatly appreciated and will directly support advancements in this field of research.

**7. Privacy and Data Use**

- Your responses will be anonymized and used solely for research purposes.
- No personally identifiable information will be collected or stored.

### 8. Questions or Concerns

If you have any questions about the study or encounter technical issues, please contact us. We are happy to assist you.

### 9. Consent

By proceeding with the study, you confirm that:

- You have read and understood the instructions.
- You agree to participate voluntarily.
- You understand that you can withdraw at any time without penalty.

Thank you again for your time and effort. Let's get started!

---

**Compensation.** Participants were informed that their participation was entirely voluntary based on their willingness to contribute to scientific research, without financial compensation. No incentives or pressures were applied to encourage involvement. Participants could withdraw at any time.

**Minimal risks.** Before beginning the study, participants were provided with detailed information about the purpose of the research, and their rights as participants. Consent was obtained from all participants prior to their involvement. The risks associated with this study were minimal. The tasks posed no physical, psychological, or emotional harm to participants. Participants were only required to perform binary voting tasks, which involved evaluating outputs generated by different methods. All interactions were conducted through a secure online platform, ensuring privacy and anonymity. No sensitive data or personally identifiable information was collected during the study.

**Approval.** This study received approval from the research institution.

## G   Societal Impacts and Safeguards

The advancements in multi-shot portrait video generation, as exemplified by our work on EchoShot, present profound societal impacts across numerous domains. By enabling high-quality, identity-consistent, and content-controllable portrait video creation, EchoShot democratizes access to advanced video production tools, empowering creators of all skill levels to produce professional-grade visual media. This innovation streamlines workflows in industries such as filmmaking, advertising, virtual avatars, and social media content creation, leading to significant improvements in efficiency, cost savings, and creative flexibility. The ability to generate consistent multi-shot videos with fine-grained control over attributes like facial expressions, outfits, and motions has the potential to revolutionize storytelling, personalized media, and digital human modeling.

However, the rapid adoption of such generative technologies also raises important challenges. Job displacement may occur for traditional video editors and artists who rely on conventional methods, necessitating reskilling and adaptation to remain competitive in an evolving job market. Ethical concerns are equally critical, as the misuse of generated videos (e.g.. creating deepfakes, impersonations, or misleading content) can undermine public trust and exacerbate misinformation. Ensuring that models like EchoShot are trained on unbiased datasets is essential to prevent the reinforcement of harmful stereotypes or biases in generated content. To address these issues, we inherit safeguards from foundational models Wan2.1, including mechanisms for detecting inappropriate or harmful outputs, while adhering to strict usage guidelines. Furthermore, by open-sourcing our models and dataset, we aim to foster transparency, encourage responsible use, and facilitate community-driven improvements in ethical generative AI development.

