# OpenReview forum: "EchoShot: Multi-Shot Portrait Video Generation"
_NeurIPS.cc/2025/Conference — NeurIPS 2025 poster_

### Official Review · Reviewer_SaBb · 2025-07-01

**Clarity:** 3
**Significance:** 3
**Originality:** 3
**Rating:** 4
**Confidence:** 4

**Summary:**

This paper introduces a framework for multi-shot portrait video generation that addresses the limitation of existing text-to-video models in maintaining identity consistency across multiple shots. Built on a diffusion transformer (DiT) architecture, the proposed method incorporates shot-aware position embedding mechanisms to model inter-shot relationships and align text-video content. The proposed method can be extended for reference-based personalized multi-shot generation and infinite shot synthesis. To train their model, the authors collect a large-scale multi-shot portrait video dataset. Experimental results demonstrate superior performance in identity consistency, prompt controllability, and visual quality compared to baselines.

**Questions:**

* Line 199: new patter -> new pattern

* It is unclear whether the provided videos in the supplemental materials are generated **with face ID condition (Sec. 3.3 PMT2V) or not**.

**Ethical Concerns:**

["NO or VERY MINOR ethics concerns only"]

**Final Justification:**

Thank the authors for the detailed rebuttal. I am glad to see that the authors have establish an open-sourcing timeline and promised to release the evaluation set as well as the visual results. Please do not just publish a coming soon link. I would like to maintain my initial positive rating after reading the rebuttal and other reviews.

**Limitations:**

The authors discussed the limitations in Line 297-299, but the discussion is too brief. I think it would be better if the authors addressed the limitations in more details in the appendix.

**Paper Formatting Concerns:**

no formatting concerns

**Quality:**

3

**Strengths And Weaknesses:**

### Strengths

* Different from existing single-shot t2v works, this submission focuses on the multi-shot portrait video generation (MT2V) task,  addressing a practical need in content creation workflows.

* The proposed method does not introduce any architecture changes to the pre-trained t2v backbone. Instead, it only replaces the continuous RoPE in current t2v models with shot-aware ones, i.e., Temporal-Cut RoPE (TcRoPE) and Temporal-Align RoPE (TaRoPE), without introducing new learnable parameters. These designs enable native multi-shot training without additional computational overhead.

* The introduction of PortraitGala (250k single-shot and 400k multi-shot clips) fills a significant void in multi-shot human-centric datasets. The authors promise to release the model and dataset, which may promotes reproducibility and community advancement.

### Weaknesses
* More video results should be provided. As a video generation work, video results are crucial for readers to evaluate the effectiveness of the proposed method. Unfortunately, the authors only provide the videos with simply 3 shots and the generation conditions are unknown (MT2V? PMT2V? or InfT2V?). I think if would facilitate better evaluation and comparison if the videos corresponding to Figure 6 and 7 are also provided and the generation setting are further clarified.

* The proposed framework seems to only suit for the single-subject setting. However, multi-shot video generation is more useful for generating scenes with multi-character communications and interactions (e.g., an interview or a sport event). It is unclear how the proposed framework would be adapted to multi-character settings.

---

> ### Author Rebuttal · Authors · 2025-07-30
>
> __We are so grateful that all the reviewers give our work positive initial ratings (4454)__, which is a tremendous encouragement for us. We particularly thank you for your insightful feedback and for highlighting the innovation of our work. For each of the questions, we provide a detailed clarification below.
>
> ***
>
> ## Q1: Video Results
> __All videos in our supplementary materials were generated under the MT2V setting__, which represents the core contribution of our work for multi-shot generation from textual descriptions. These videos are representative samples from our 100-prompt evaluation set, which is the primary basis for the quantitative and qualitative analysis in our paper. The prompts, featuring 2 to 4 shots, were generated by LLM (Gemini-2.5 Pro) to ensure diversity and reduce bias.
>
> We would like to __provide further details about the inference of two extensions__. Throughout the inference process, the resolution is 480p and the total length is 8 seconds, the same as MT2V. The PMT2V and InfT2V models are tuned based on the MT2V model. For PMT2V inference of Figure 6, given an input image, we first instruct Gemini-2.5 Pro to describe the ethnicity, gender, age group of the person and generate multi-shot prompts based on the attributes. We then randomly decide a segmentation of shot lengths and crop the face region with a face detection model. Finally, we input the face crop, the shot-length segmentation and multi-shot prompts to PMT2V model to generate the personalized multi-shot video. For InfT2V inference of Figure 7, we instruct Gemini-2.5 Pro to generate 10 prompts ($p_1$-$p_{10}$) featuring the same person and sample 10 random noises $z_1$-$z_{10}$. $p_1, z_1$ is treated as the reference. We first input $z_1, z_2$ and $p_1, p_2$ to generate shots $s_1$ and $s_2$, using the InfT2V model. Then, we iteratively input $z_1, z_k$ and $p_1, p_k$ ($k\in \[3, 10\]$) to generate the other eight shots.
>
> Unfortunately, the rebuttal policy prevents us from supplying additional video files of Figures 6 and 7. Instead, we __provide partial prompts used for Figures 6 and 7 (please see postscript)__. Besides, to enhance reproducibility and contribute to the community, __we promise to publicly release our full evaluation set, the detailed evaluation methodology, and the visual results from EchoShot and all baseline models__. We have established a clear open-sourcing timeline. In fact, to facilitate early access, we have already publicly released codes and weights. We believe this will significantly strengthen our work and be a valuable resource for future research.
>
> ## Q2: Extension to Multi-Character
> This is an excellent question that we have also discussed ourselves. Rethinking EchoShot, it is __inherently adaptable to general multi-shot scenarios including single character, multiple characters and even objects__. The consistency across shots can be captured by introducing specific semantic patterns as indicators. Then EchoShot can automatically learn the correspondence between these semantics and the generated visual content. For your consideration, here we propose one potential solution for the multi-character scenario: __annotate each character in text prompts with a unique identifier__. For instance:
>
> > [Shot 1] (PERSON 1: a man with black hair) is running in a park …
>
> > [Shot 2] (PERSON 1: a man with black hair) is walking on the beach with (PERSON 2: a young boy wearing a blue shirt) ...
>
> > [Shot 3] (PERSON 2: a young boy wearing a blue shirt) is talking to (PERSON 3: a woman in a white dress) ...
>
> Then EchoShot could be trained directly on a dataset structured this way. We consider this a very promising direction for future work.
>
> ## Q3: Typos
> We will perform a thorough proofread of the entire manuscript to correct these and any other errors.
>
> ***
>
> Thanks again for your efforts. We believe the rebuttal stage is highly meaningful, and we will add the rebuttal discussion to the new version of our paper as much as possible. We would be greatly grateful if you would consider raising your score in light of these responses.
>
> ***
>
> ### Postscript 1: Prompts of Figure 6
>
> Morgan Freeman
> > [Shot 1] This segment shows a black, male, elderly person. The character's hairstyle is short, grey hair. The character is wearing a knitted cardigan, paired with a shirt and a tie. The character's expression is serious, with focused eyes, as if they are listening attentively to the phone. The corners of the character's mouth slightly downward, revealing a contemplative emotion. The character's movements are sitting on a beige patterned sofa, holding a white vintage telephone receiver in their right hand and holding an open notebook in their left hand. The video scene is a luxuriously decorated room with dark brown wooden walls as the background. There is a table on the left with a white tablecloth. The video is lit in a low-key warm tone, with light mainly focused on the character's face and a relatively dark background, creating a mysterious and oppressive atmosphere.
>
> > [Shot 2] This segment shows a black, male, elderly person. The character's hairstyle is short, grey hair. The character is wearing a knitted cardigan, paired with a shirt and a tie. The character’s expression is relaxed and cheerful. The character's movement is sitting on a rocking chair, with their left hand gently resting on the armrest. The video scene is a sunlit porch with lush green grass and distant mountains in the background, creating a beautiful scenery. The video is illuminated with bright light, making the facial details and background of the characters clearly visible.
>
> > [Shot 3] This segment shows a black, male, elderly person. The character's hairstyle is short, grey hair. The character is wearing a knitted cardigan, paired with a shirt and a tie. The character’s expression is serene and calm, with a kind gaze and a gentle smile at the corner of their mouth, conveying a sense of tranquility and wisdom. The character's action is resting on a bench in the park, with his eyes closed and listening to the sounds around them. The video scene is a vibrant park with dense trees in the background. The video is illuminated with bright natural light, evenly illuminating the characters and surrounding environment, creating a relaxed and pleasant atmosphere.
>
> Emma Watson
> > [Shot 1] This segment shows a Caucasian, female, young person. The character's hairstyle is shoulder length short hair, with a light golden color and a black thin headband. The hair is fluffy, slightly messy, and has a slight inward button at the end. The character is wearing a light blue shirt. The character's expression is happy and joyful. The character's movements are standing still, raising the head, looking up at the top of the screen and speaking. The video scene is a dark forest. There are tall trees in the background with mist and trees. The ground is covered with fallen leaves and some green plants. The overall lighting is dim, creating a gloomy and mysterious atmosphere. The light mainly comes from above, illuminating the face of the character.
>
> > [Shot 2] This segment shows a Caucasian, female, young person. The character's hairstyle is shoulder length short hair, with a light golden color and a black thin headband. The hair is fluffy, slightly messy, and has a slight inward button at the end. The character is wearing a deep red short sleeved dress and gold wire glasses. The collar of the dress is circular, and the sleeves reach the middle of the upper arm. The character's expression is slightly worried and uneasy. The character's movements are standing still and gazing downward. The video scene is a desolate place shrouded in mist. In the background, you can see some blurry mountain and forest. On the right side of the character, there is a tall stone pillar with a rough surface and some blurry patterns engraved on it. The lighting in the video is low-key, with overall dim lighting, creating a oppressive and mysterious atmosphere. The light mainly comes from above and in front, but the light is soft and there are no obvious shadows.
>
> > [Shot 3] This segment shows a Caucasian, female, young person. The character's hairstyle is shoulder length short hair, with a light golden color and a black thin headband. The hair is fluffy, slightly messy, and has a slight inward button at the end. The character is wearing a black V-neck dress. The character's expression is extremely sad and painful, with tears in their eyes, red eyes, and slightly parted lips. Their overall mood is low. The character's movements are standing still, with their head slightly raised, their eyes looking up, their lips slightly trembling. The video scene is indoors, with a light brown wall adorned with abstract tree patterns. The overall scene is simple and artistic. The video is illuminated with soft light, which is uniform and bright.
>
> ### Postscript 2: Prompts of Figure 7
>
> > [Shot 1] This segment shows an Asian, female, and young person. The character's hairstyle is dark brown, with split hair and two thin braids tied at the back of the head. The ends of the braids are secured with black rubber bands. The character is a white high-neck shirt layered under a beige trench coat. The character’s expression is gentle and pleasant. The character is sitting and is in the process of drinking from a white ceramic coffee cup. The video scene is in a cafe, with a softly blurred background showing wooden furniture and green plants. The lighting is bright and soft, illuminating the subject evenly and creating a warm, inviting atmosphere.
>
> > [Shot 2] This segment shows an Asian, female, and young person. The character's hairstyle is dark brown, with split hair and two thin braids tied at the back of the head. The ends of the braids are secured with black rubber bands. The character is wearing a white t-shirt and a silver chain necklace, layered under a blue denim jacket ...
>
> （Omitted due to text length limitations）

---

> > ### Comment · Reviewer_SaBb · 2025-08-07
> >
> > Thank the authors for the detailed rebuttal. I am glad to see that the authors have establish an open-sourcing timeline and promised to release the evaluation set as well as the visual results. Please do not just publish a coming soon link. I would like to maintain my initial positive rating after reading the rebuttal and other reviews.

---

> > > ### Author Response · Authors · 2025-08-08
> > >
> > > We are deeply appreciative of your recommendation for acceptance and your valuable input in enhancing this work. We will continue to advance both the development and open-sourcing of this project, as we have promised.

---

### Official Review · Reviewer_44zS · 2025-07-01

**Clarity:** 4
**Significance:** 4
**Originality:** 3
**Rating:** 5
**Confidence:** 4

**Summary:**

The authors of this paper propose EchoShot, a diffusion-based portrait video generation method capable of generating multiple shots with consistent identity. Videos are generated from text prompts and can be further personalized based on the appearance encoded in a facial image. The authors address two tasks: PMT2V and InfT2F. The objective of PMT2V is to personalize the generated shots using ID information encoded in a facial image. The objective of InfT2F is to generate an infinite number of shots with the same identity. The authors propose fixing the prompts and generation of the first shot while varying the prompts of the remaining shots. They propose RefAttn, which enables inter-shot attention while fixing the first shot to ensure ID consistency. The authors also propose modified RoPE position embeddings (TcRoPE and TaRoPE) to enable multi-shot training. Their method is trained on a new dataset, PortraitGala, which comprises multi-shot video clips of the same identity and is thus well-suited for this purpose. Furthermore, the authors plan to release the dataset, which will be a significant benefit to the community.  Their method generates impressive results, surpassing SOTA methods in identity consistency, prompt controllability, and visual quality. Additionally, a user study highlights that EchoShot better aligns with user preferences than SOTA methods.

**Questions:**

- What is the runtime for generating videos?
- How do you select the face image for PMT2V during training?
- Why does the generated face in Figure 6 (right block) not match the appearance of the input face? Ideally, only the hairstyle should change based on the text prompt.

**Ethical Concerns:**

["NO or VERY MINOR ethics concerns only"]

**Final Justification:**

The authors addressed all points to my satisfaction. Therefore, I continue to support the acceptance of the paper and my positive rating remains unchanged.

**Limitations:**

The authors discussed limitations but not potential negative societal impacts.

**Paper Formatting Concerns:**

No Concerns.

**Quality:**

4

**Strengths And Weaknesses:**

Strengths:

- The paper is well written and meets the standards of NeurIPS.
- The method outperforms state-of-the-art video diffusion models for portrait video generation.
- The proposed method allows for the generation of multi-shot videos with excellent ID consistency, which is highly valuable.
- The method achieves excellent ID consistency across different shots generated of the same person.
- The dataset will be made publicly available, which is a significant contribution because it is designed to improve cross-shot identity consistency.
- The results look very natural.
- The user study supports the authors' claims.


Weaknesses:

- To achieve the desired cross-shot ID consistency, a replicate of the first shot must be generated in every inference attempt.
- The generated appearance does not match the face input when the hairstyle in the text prompt differs from the hairstyle in the face image. In the right block of Figure 6, EchoShot does not generate shots that align with the person's overall appearance in the face image. In this particular example, I would expect only the hairstyle to be altered. Although the generated appearance of the multiple shots seems consistent, it does not match the input face image. In comparison, the hairstyle prompt matches the face input in the left block, leading to generated shots that match the face input image.


Since there are no video results, I cannot judge the temporal consistency or realism of the generated videos.

---

> ### Author Rebuttal · Authors · 2025-07-30
>
> __We are so grateful that all the reviewers give our work positive initial ratings (4454)__, which is a tremendous encouragement for us. We particularly thank you for your recognition and for highlighting the overall contributions of our work. For each of the questions, we provide a detailed clarification below.
>
> ***
>
> ## Q1: Inference Runtime
> The inference time of EchoShot is __identical to the original Wan-1.3B model__, because our proposed RoPE design introduces no additional computational overhead. To specify, generating a multi-shot video of __8 seconds in total length at 480p resolution__ requires __24GB of VRAM__ and takes approximately __360 seconds__ on a single NVIDIA A100 GPU.
>
> ## Q2: Faces for PMT2V Training
> We assume you are asking about __the construction process of our PMT2V dataset__. For each video clip, we employ the following multi-stage process to obtain the face:
> - We use a face detection model to extract face crops from every frame.
> - These crops undergo a rigorous filtering process based on multiple criteria, including face orientation, sharpness, and the relative size of the face within the frame. The filter is driven by in-house analysis tools.
> - We randomly sample at most 5 face images from the filtered crops, ensuring a minimum temporal gap of 1 second between any 2 face images.
>
> ## Q3: Face Consistency
> Regarding the Emma Watson example in Figure 6, the perceived imperfect match between the generation and the input can be attributed to both objective and subjective factors.
> - __Objective Factor: The Inherent Trade-off between Identity and Motion__. There is a common challenge in personalization tasks involving the trade-off between preserving identity and generating dynamic motion. A single reference image input provides limited information, making it __difficult to perfectly model the full spectrum of a person's facial dynamics__. Our model, trained on a diverse dataset, is designed to excel at generating a wide range of dynamic features, such as varied expressions and head poses. In the Emma Watson example, EchoShot generates complex facial dynamics (e.g., laughing, crying) while baselines fail to achieve. This strong emphasis on motion can __lead to subtle deviations from the static input image__, which is perceived as an imperfect identity match.
> - __Subjective Factor: Perceptual Sensitivity to Celebrity Faces__. Human perception is more sensitive to inconsistencies in the faces of well-known celebrities. This is because we have strong prior knowledge of their appearance from various angles and in diverse contexts. We once conducted a small-scale comparative user study, where we provided videos generated by EchoShot based on celebrity faces and non-celebrity random faces. We found that __participants generally perceived a higher degree of identity consistency when seeing generations of non-celebrity faces__.
>
> We believe this challenge can be effectively mitigated by incorporating multiple reference images to provide a more comprehensive identity representation. We consider this a promising direction for future work.
>
> ## Q4: Video Result
> We included video examples of EchoShot __in the supplementary materials (zip file)__. We believe these results effectively demonstrate the strong temporal consistency and excellent realism achieved by our model. Besides, __we promise to publicly release our full evaluation set and the visual results from EchoShot and all baseline models__. Our goal is to establish the first comprehensive benchmark for multi-shot portrait video generation, which we believe will significantly strengthen our work and be a valuable resource for future research.
>
> ***
>
> Thanks again for your efforts. We believe the rebuttal stage is highly meaningful, and we will add the rebuttal discussion to the new version of our paper as much as possible.

---

> > ### Comment · Reviewer_44zS · 2025-08-04
> >
> > Thank you for your rebuttal and clarifications.
> >
> > > Inference Runtime
> >
> > How does this compare to related methods?
> >
> > > Faces for PMT2V Training
> >
> > I wanted to know how you select the face image used for conditioning during training.
> >
> > > Faces for PMT2V Training
> >
> > I agree that your method better generates facial dynamics. However, the identity variation in this example is real and does not result from visual bias. I have seen face reenactment methods that model complex facial dynamics while achieving nearly perfect identity consistency. If your prompts closely matched Emma Watson's appearance in this example, I believe the ID consistency would be significantly better, like in the other example in Figure 6. As I said, it seems like your method prioritizes generating videos with matching text prompts. I think this could be improved in future work.
> >
> > > Video Result
> >
> > Thank you for the clarification!

---

> > > ### Author Response · Authors · 2025-08-05
> > >
> > > Thank you for your follow-up response. We would like to provide additional clarifications.
> > > ***
> > > > Inference Runtime
> > >
> > > We provide a comparison between EchoShot and __all open-source baseline methods__ mentioned in the paper. All models are evaluated on a single NVIDIA A100 using default settings including hyper-parameter, resolution, length, etc. Note that __all baselines produce single-shot videos__ in this comparison and generating multi-shot outputs would exponentially increase their inference time.
> > >
> > > | Model | Width x Height x Frame | Peak GPU Memory | Total Inference Time |
> > > |---| --- | --- |---|
> > > | EchoShot (Wan-1.3B) | 832 x 480 x 125 (multi-shot) | 24GB | 360s |
> > > | StoryDiffusion + Wan-14B | 832 x 480 x 81 (single-shot) | 41GB | 8s + 810s |
> > > | IC-LoRA + Wan-14B | 832 x 480 x 81 (single-shot) | 41GB | 23s + 810s |
> > > | ConsisID | 720 x 480 x 49 (single-shot) | 44GB | 214s |
> > >
> > > > Faces for PMT2V Training
> > >
> > > We provided details on PMT2V dataset construction in the rebuttal text. Based on our understanding of your follow-up feedback, we would like to further explain how the dataset is sampled during training here. (If we misunderstood, please let us know. We would be grateful if you could provide a more detailed description of the question.) To specify, we train the PMT2V model with __"cross-pair" conditioning data__ to enhance controllability and diversity, following early works (e.g. ConsisID). For example, given 6 video clips of the same person in a training step, we first randomly sample 4 clips and concatenate them into a 4-shot video. We then randomly sample a face from the remaining 2 clips as the face condition. Thus, __the input face is not included in the 4-shot video__, forming a cross pair. The face, the 4-shot video and the caption are input to the model for training. We hope our understanding of your query is accurate and that the provided explanation is of assistance.
> > >
> > > > Face Consistency
> > >
> > > We appreciate the feedback and will incorporate this limitation into the manuscript, highlighting it as a potential avenue for future research.
> > > ***
> > > Thanks again for your response. The communication with you is enjoyable and inspiring. Please let us know If you have any further questions.

---

> > > > ### Comment · Reviewer_44zS · 2025-08-06
> > > >
> > > > Thank you for the explanations. The author satisfactorily answered my questions. I stand by my decision to accept the paper.

---

> > > > > ### Author Response · Authors · 2025-08-06
> > > > >
> > > > > We are sincerely grateful for your recommendation for acceptance as well as your great contributions to improving this work, and we enjoyed participating in such a constructive discussion. We will continue to advance the development and open-sourcing of this work, as we have promised.

---

### Official Review · Reviewer_znTz · 2025-07-02

**Clarity:** 3
**Significance:** 3
**Originality:** 3
**Rating:** 4
**Confidence:** 3

**Summary:**

The paper introduces a novel task of multi-shot portrait video generation where the portrait of a single person is consistent identity-wise across all of the shots.

Contributions are as follows:

(1) PortraitGala, a large-scale portrait dataset comprising 250k one-to-one single-shot video clips and 400k many-to-one multi-shot video clips.

(2) A new position embedding mechanism termed Temporal-Cut RoPE (TcRoPE) is introduced to resolve challenges in variable shot counts, flexible shot durations, and inter-shot token discontinuities.

(3) Temporal-Align RoPE (TaRoPE) (in the video-text cross-attn module) is introduced for better text control.

(4) An external face encoder enables reference-image based personalized multi-shot video generation (PMT2V).

(5) a disentangled RefAttn mechanism is used for infinite shots video generation (InfT2V).

**Questions:**

I would like to know (1) if the proposed method can generalize to videos that contain non-photorealistic characters, (2) what are the limitations considering camera view/rotation, (3) is there a way for the method to be extended to multiple people in one shot or to generate videos with multiple persons where at least one of them is consistent across the shots?

(4) Generated videos (supmat) seem static and the quality in the eyes and teeth regions is low. Also, it seems there are issues with the overall image quality in the first few frames. Why?

(5) Lines 42/43: Can the authors explain "the bottleneck" effect in more detail?

**Ethical Concerns:**

["NO or VERY MINOR ethics concerns only"]

**Final Justification:**

I would like to thank the authors for the rebuttal. I think that this work is interesting, well evaluated, and the dataset contribution seems significant. I also appreciate that the authors have mentioned that the method can generalize to certain out-of-the-distribution domains and noted the limitations/future directions. My main concern with the current version remains the issues of video quality consistency and artifacts noted earlier. For this reason, I am keeping my original rating, but given the strengths of the work, I recommend acceptance as a poster.

**Limitations:**

yes

**Paper Formatting Concerns:**

/

**Quality:**

3

**Strengths And Weaknesses:**

Strengths:

The problem is interesting. Generating multiple videos of the same person with fine-grained control is important for applications in the emerging creative AI filmmaking industry.

The method is evaluated well, especially considering qualitative and quantitative comparisons with the baselines. There are also additional results and ablation studies in the appendix.

Dataset contribution seems significant.

Paper is well written.


Weaknesses:

Lines 42/43: Can the authors explain "the bottleneck" effect in more detail?

Can the camera rotate around the person (360 degrees) or is the method limited to frontal shots only?

Can the framework be extended to multiple people?

Does the method work for photorealistic examples only or does it also generalize to out-of-distribution domains?

Generated videos (supmat) seem static and the quality in the eyes and teeth regions is low. Also, it seems there are issues with the overall image quality in the first few frames. Why?

I would suggest adding more important ablation studies (model) to the main paper.

It would have been helpful to see all of the videos that were used in the user study.

There are typos in some parts:

- Section 4: Experiment -> Experiments

- Figure 6: “not follow” -> doesn’t follow

---

> ### Author Rebuttal · Authors · 2025-07-30
>
> __We are so grateful that all the reviewers give our work positive initial ratings (4454)__, which is a tremendous encouragement for us. We particularly thank you for the meticulous feedback and for highlighting the significance of our work. For each of the questions, we provide a detailed clarification below.
>
> ***
>
> ## Q1: Generalization to Non-Realistic Styles
> Though our training dataset predominantly consists of videos in a realistic style, we have observed that EchoShot exhibits __an emergent capability to generate consistent shots in certain artistic styles__, such as 3D animation. We are sorry that due to rebuttal policy, we cannot include the corresponding video examples. To explain this phenomenon, we hypothesize that EchoShot effectively __inherits the versatile generative capabilities of the base mode__ (i.e., Wan-1.3B), benefiting from our proposed RoPE designs. However, due to the lack of explicit training on these artistic domains, the visual quality of these stylized generations is not yet on par with that of the realistic style. Furthermore, __for styles that are beyond the capability of the base model itself__, such as stop-motion animation, EchoShot is consequently unable to generalize.
>
> ## Q2: Camera Control
> Our dataset annotation and model generation capabilities focus on __common camera movements__, including static, handheld, pan left/right, tilt up/down, zoom in/out, limited-angle orbiting shot. Consequently, the model currently __does not support more complex camera motions__, such as 360-degree orbiting shots or fully customized camera paths. We acknowledge this as a current limitation and a valuable direction for future research.
>
> ## Q3: Extension to Multi-Character
> This is an excellent question that we have also discussed ourselves. Rethinking EchoShot, it is __inherently adaptable to general multi-shot scenarios including single character, multiple characters and even objects__. The consistency across shots can be captured by introducing specific semantic patterns as indicators. Then EchoShot can automatically learn the correspondence between these semantics and the generated visual content. For your consideration, here we propose one potential solution for the multi-character scenario: __annotate each character in text prompts with a unique identifier__. For instance:
>
> > [Shot 1] (PERSON 1: a man with black hair) is running in a park …
>
> > [Shot 2] (PERSON 1: a man with black hair) is walking on the beach with (PERSON 2: a young boy wearing a blue shirt) ...
>
> > [Shot 3] (PERSON 2: a young boy wearing a blue shirt) is talking to (PERSON 3: a woman in a white dress) ...
>
> Then EchoShot could be trained directly on a dataset structured this way. We consider this a very promising direction for future work.
>
> ## Q4: Video Quality
> Artifacts do occasionally occur during generation, which __we attribute to the limited parameters of the base model__. Wan-1.3B is a very small model compared to peer models. Our internal tests confirm that __original Wan-1.3B exhibits inferior performance to original Wan-14B in terms of dynamic quality and face details__. Considering the computation cost, we decided to conduct this study on Wan-1.3B. Though, the method we proposed can be seamlessly applied to larger models. For future work, we plan to __explore parameter-efficient fine-tuning methods__ (e.g., LoRA) to adapt EchoShot to the 14B model at a lower cost. Based on the current results, we are confident that migrating to a more powerful base model would achieve even higher quality.
>
> ## Q5: Explanation of the Bottleneck Effect
> We use the term *the bottleneck effect* to describe a phenomenon in *keyframe-to-video synthesis pipelines*. These cascaded pipelines typically consist of a consistent text-to-image (CT2I) model followed by an image-to-video (I2V) model, where the output of the former serves as the input for the latter. In this architecture, the quality of the final video generated by the I2V model is heavily constrained by the quality of the initial keyframes produced by the CT2I model. Thus, __the performance of the CT2I model acts as a "bottleneck”, limiting the overall system's potential, regardless of the I2V model's inherent capabilities__. This effect is empirically validated in Figure 5 of our paper. Both "StoryDiffusion+Wan" and "IC-LoRA+Wan" use the same I2V model (Wan-14B-I2V) and different CT2I models (StoryDiffusion and IC-LoRA), yet their resulting visual quality differs dramatically. This disparity arises because IC-LoRA, as a CT2I model, exhibits superior overall performance compared to StoryDiffusion. In this case, the capacity of Wan-14B-I2V is underutilized.
>
> ## Q6: Ablation Study
> Thanks for your advice. We will revise the paper to add critical ablation studies into the main part to strengthen our experimental analysis.
>
> ## Q7: User Study Videos
> The supplementary material (zip file) contains video examples generated by EchoShot for user study. We apologize that due to the policy, we are unable to provide videos generated by baseline methods in this rebuttal. To enhance reproducibility and contribute to the community, __we promise to publicly release our full evaluation set, the detailed evaluation methodology, and the visual results from EchoShot and all baseline models__. Our goal is to establish the first comprehensive benchmark for multi-shot portrait video generation, which we believe will significantly strengthen our work and be a valuable resource for future research.
>
> ## Q8: Typos
> We will perform a thorough proofread of the entire manuscript to correct these and any other errors.
>
> ***
>
> Thanks again for your efforts. We believe the rebuttal stage is highly meaningful, and we will add the rebuttal discussion to the new version of our paper as much as possible. We would be greatly grateful if you would consider raising your score in light of these responses.

---

> > ### Comment · Reviewer_znTz · 2025-08-05
> >
> > Thank you for the rebuttal. My current rating is between a (4) and a (5), primarily due to (1) the lower video quality in the current version, and (2) the lack of video results, as also noted by Reviewer SaBb. I will make a final decision during the next phase.

---

> > > ### Author Response · Authors · 2025-08-06
> > >
> > > Thank you for the follow-up response and for your recommendation of acceptance. We truly appreciate the insightful and constructive discussion with you, which is instrumental in refining our work. We will continue to advance the development and open-sourcing of this work, as we have promised.

---

### Official Review · Reviewer_Ttb3 · 2025-07-02

**Clarity:** 3
**Significance:** 2
**Originality:** 3
**Rating:** 5
**Confidence:** 4

**Summary:**

EchoShot is a system for multi-shot portrait video generation with subject-identity consistency and shot-wise controllability. Built on a 1.3 B-parameter Wan-DiT backbone, the framework introduces two shot-aware rotary-position-embedding variants: Temporal-Cut RoPE (TcRoPE) for marking inter-shot boundaries in self-attention and Temporal-Align RoPE (TaRoPE) for suppressing cross-shot leakage in text-video cross-attention.Training is driven by PortraitGala, a newly curated 600 k-clip dataset (400 k identities, 1 k h video) with fine-grained captions Three tasks are demonstrated:
MT2V - text-to-multi-shot video;
PMT2V - personalised multi-shot generation from a face reference;
InfT2V - unbounded shot extension via a disentangled RefAttn.

On a private 100-prompt benchmark,  EchoShot improves identity consistency and prompt alignment over keyframe pipelines and HunyuanVideo, while users prefer its outputs in >70 % pairwise votes. All code and data are promised for open-source release upon acceptance

**Questions:**

1. Full release of the evaluation set and outputs.
Will you publish the entire 100-prompt test set together with your inference videos and metric scripts?
A public release before the rebuttal deadline would let reviewers and future work verify your claims.

2. Concrete timeline for open sourcing code, weights and PortraitGala.
Please specify an exact date (e.g., “on or before 1 October 2025”) for releasing training code, inference checkpoints and the PortraitGala dataset (or a legally shareable subset).
A firm commitment would resolve reproducibility concerns.

3. Public benchmarks. Could you run EchoShot and baselines on an open set (e.g., a 20-clip WebVid subset) and publish the numbers? Demonstrating similar gains outside your data would markedly strengthen the work.

Based on author's answers, I'm willing to either increase or decrease my scores.

**Ethical Concerns:**

["NO or VERY MINOR ethics concerns only"]

**Final Justification:**

I am satisfied with the author's response and will advocate for the acceptance of this paper.

**Limitations:**

The paper notes that EchoShot cannot yet extend an existing shot into longer continuous footage. It should also spell out potential misuse for non-consensual identity swaps and describe intended safety mechanisms; an explicit plan would be welcome.

**Paper Formatting Concerns:**

No concerns

**Quality:**

3

**Strengths And Weaknesses:**

Strengths:
Substantial gains in automatic metrics and a user study confirm that TcRoPE + TaRoPE preserve identity across shots and respect fine-grained prompts. The idea of embedding shot indices directly into 3-D and 1-D RoPE is conceptually simple yet addresses variable-length shots and cross-shot prompt bleed-over in a unified way.
Architecture diagrams, loss formulas and hyper-parameters are clearly documented; compute cost (3 500 A100 GPU-h) is disclosed
Multi-shot portrait generation is a frequent real-world request (story-boarding, virtual try-on, character vlogging). By training natively on many-to-one clips, EchoShot sets a useful precedent beyond iterative keyframe hacks.

Weaknesses:
All metrics rely on an unreleased 100-prompt set; no numbers are reported on public corpora. Reproducibility therefore hinges on a future release.
Limited baseline breadth. Comparisons omit recent multi-event generators and restrict closed-source Kling to PMT2V only, leaving the competitive landscape unclear.
Conditional openness. Code, checkpoints and PortraitGala will be released “upon acceptance”, not at submission, delaying external validation.

---

> ### Author Rebuttal · Authors · 2025-07-30
>
> __We are so grateful that all the reviewers give our work positive initial ratings (4454)__, which is a tremendous encouragement for us. We particularly thank you for the insightful feedback and for highlighting the strengths of our work. Our primary motivation is to contribute to the video generation community, so we are willing to take any action that enhances the openness and impact of our work. For each of the questions, we provide a detailed clarification below.
>
> ---
>
> ## Q1: Evaluation Set
> Thank you for highlighting the importance of an open evaluation set. We agree that this is crucial for enabling future work to compare against and build upon our method. Thus, __we promise to release our full evaluation set, video results and evaluation pipeline__.
>
> To provide immediate clarity on our evaluation process, we would like to __share some key details here__. The Vision-Language Model (VLM) used for our automated scoring is Gemini-2.5 Pro. For the *Prompt Controllability* metrics, the VLM scores the generated videos across nine distinct aspects: ethnicity, gender, age group, hairstyle, apparel, expression, action, background, and lighting. The final scores are calculated as follows: *App. score*: the average of scores for ethnicity, gender, age group, hairstyle, and apparel. *Mot. score*: the average of scores for expression and action. *Bg. score*: the average of scores for background and lighting. For the *Visual Quality* metrics, the *Sta. score* and *Dyn. score* are directly scored by the VLM.
>
> Due to the format limitations of the rebuttal, we can only __provide a prompt example from our evaluation set and the scoring instructions input to Gemini-2.5 Pro (please see the postscript)__. The full benchmark will be released according to the timeline below.
>
> ## Q2: Timeline for Open-Sourcing
> We have established a clear open-sourcing timeline. In fact, to facilitate early access, we have already finished certain steps:
>
> - Before June 15th, 2025: Launch project page and generation gallery. __(Finished)__
> - Before July 15th, 2025: Publicly release codes. __(Finished)__
> - Before July 15th, 2025: Publicly release the model weight. __(Finished)__
> - Before August 15th, 2025: Publicly release the full test set, evaluation details, and baseline benchmarks.
> - Before September 30th, 2025: Publicly release the PortraitGala dataset.
>
> We are committed to this schedule and believe it directly addresses the concerns regarding reproducibility.
>
> ## Q3: Evaluation on Public Benchmarks
> The primary reason for not including evaluation on public benchmarks was __the lack of a public a large-scale dataset with many-to-one annotations__ (i.e., multiple video shots of the same person paired with consistent text descriptions), to the best of our knowledge. This very gap is exactly the reason why we construct and commit to releasing the PortraitGala dataset. Regarding your specific suggestion of WebVid, it consists of diverse scenes and is designed for general-purpose video generation. It __lacks sufficient videos and annotations of the same person in different contexts__. As a result, we are unable to conduct training or evaluation on WebVid in the multi-shot portrait generation setting. For this issue, __we promise to release PortraitGala dataset, detailed evaluation methodology, and the videos generated by EchoShot and all baseline models__. By doing so, we will establish the first public benchmark specifically for the multi-shot portrait generation task. We believe this contribution will markedly strengthen our work, provide a lasting resource for the community, and address your concerns about comparative evaluation.
>
> ---
>
> We believe the rebuttal stage is highly meaningful, and we will add the rebuttal discussion to the new version of our paper as much as possible. We would be very grateful if you would consider raising your score in light of these clarifications.
>
> ---
>
> ### Postscript 1: An Example of 100-prompt Evaluation Set
> > [Shot 1] Camera pans left, medium shot. This segment shows a Caucasian, male, middle-aged person. The character's hair is dark brown, medium length, voluminous, slightly fluffy, with natural waves, and some hair falls over the forehead and sides of the face. The character is wearing a light blue long-sleeved shirt, with the collar open, and a dark blue striped tie. The character's expression is serious, with a focused gaze, corners of the mouth slightly downturned, seemingly thinking or observing. The character's action is standing, body slightly angled to the left, head turned to the right, seemingly observing or listening, with hands naturally hanging down and no obvious body movements. The video scene is an office, with a light green background wall. On the left side, there is a whiteboard covered with names and school names, and on the right wall, a white bulletin board is hanging. The video lighting is generally dim, with light coming from the left, casting shadows on the character's face, creating a serious and oppressive atmosphere.
>
> > [Shot 2] Still shot, close-up. This segment shows a Caucasian, male, middle-aged person. The character's hair is dark brown, slightly long, layered, naturally combed back, with some hair falling on both sides of the forehead. The character is wearing a light blue and white striped shirt, with the collar open, revealing the collarbone. The character is wearing a dark blue tie with white polka dots. The character's expression is calm, slightly serious. The character's action is sitting on a sofa, body slightly leaning forward. The character occasionally makes gestures with both hands. The video scene is indoors, with a light green wall in the background, which has subtle textures. The scene is simply arranged, without excessive decoration. The video lighting is soft natural light, with light coming from the front, illuminating the character's face and upper body. The light is evenly distributed, with no obvious shadows. The overall lighting conditions are relatively bright.
>
> > [Shot 3] Still shot, close-up. This segment shows a Caucasian, male, middle-aged person. The character's hair is dark brown, slightly fluffy, naturally combed back, with slightly longer sideburns. The character is wearing a light blue Hawaiian shirt, with white and dark blue patterns printed on it, and the collar slightly open. The character's expression is relaxed and slightly smiling, with a focused gaze, seemingly listening or thinking. Occasionally, the corners of the mouth subtly turn up, showing a sense of calmness and confidence. The character's action is slight head movements, subtle lip movements, seemingly speaking, with the body remaining relatively still, shoulders relaxed, and hands possibly placed on a table or in front of the body. The video scene is an indoor office, with a blurred background. Some stacked files and office supplies are visible. The video lighting is soft natural light, with light coming from the front, illuminating the character's face. Shadows are not obvious, and the overall light is even, creating a comfortable and relaxed atmosphere.
>
> ### Postscript 2: Rating Instruction for Prompt Controllability Metrics
> > Please play the role of a video generation model evaluator. I will give you a prompt and a corresponding generated video clip. Please judge whether the content of the video clip meets the description of the prompt in terms of ethnicity, gender, age group, hairstyle, apparel, expression, action, background, and lighting. If it fully meets the description, output 100, if it does not meet the description, output 0, and if it partially meets the description, give a score between 0 and 100 according to the degree. Each aspect should be judged independently, and the entire content of the prompt must be strictly considered. Only give the results, written in the form of a python dictionary, with the key being the evaluation aspect and the value being the score.
>
> ### Postscript 3: Rating Instruction for Visual Quality Metrics
> > Please play the role of a video generation model evaluator. I will give you a prompt and the corresponding generated video clip. Please rate the video, which is divided into two scores, static and dynamic, each from 0 to 100. The static dimension (single-frame picture quality) is considered from the aspects of clarity and sharpness, color performance, detail restoration ability, noise and artifact control, composition and aesthetics, resolution consistency, etc. The dynamic dimension (continuous picture performance) is considered from the aspects of motion smoothness, motion blur processing, timing consistency, dynamic range and brightness changes, action realism, scene transition and occlusion processing, dynamic detail retention, etc. Only give the results, written in the form of a python dictionary, with the key being the evaluation aspect and the value being the score.

---

### Decision · Program_Chairs · 2025-09-17

**Decision:**

Accept (poster)

**Comment:**

This paper receives positive scores. The reviewers raise some concerns regarding the limited baseline breadth, the lack of evaluation on public benchmarks, the lack of model ablation studies, the model generalization ability to videos with non-photorealistic characters, etc. Most of the questions are addressed in the rebuttal and recognized by reviewers, with minor concerns in the issues of video temporal consistency. Authors should incorporate these review feedbacks, and release the code and demo videos in final version. Eventually, this paper is recommended to be accepted.